



# Hydrochloric acid emission dominates inorganic aerosol formation from ammonia in the Indo-Gangetic Plain during winter

Pooja V. Pawar[1,6], Sachin D. Ghude[1], Gaurav Govardhan[1], Prodip Acharja[1], Rachana Kulkarni[2], Rajesh Kumar[3], Baerbel Sinha[4], Vinayak Sinha[4], Chinmay Jena[5], Preeti Gunwani[1], Tapan Kumar Adhya[6], Eiko Nemitz[7], and Mark A. Sutton[7]

[1]Indian Institute of Tropical Meteorology (IITM), Ministry of Earth Sciences, Pune, India
[2]Savitribai Phule Pune University, Pune, India
[3]National Center for Atmospheric Research (NCAR), Boulder, CO, USA
[4]Department of Earth and Environmental Sciences, Indian Institute of Science Education and Research Mohali, Punjab, India
[5]India Meteorological Department (IMD), Ministry of Earth Sciences, Lodhi Road, New Delhi, India
[6]Kalinga Institute of Industrial Technology (KIIT), Bhubaneshwar, India
[7]UK Centre for Ecology & Hydrology (UKCEH), Edinburgh, UK

*Correspondence to*: Sachin D. Ghude (sachinghude@tropmet.res.in)

**Abstract.** The Winter Fog Experiment (WiFEX) was an intensive field campaign conducted at Indira Gandhi International Airport (IGIA) Delhi, India, in the Indo-Gangetic Plain during the winter of 2017-2018. Here, we report the first comparison in South Asia of the high temporal resolution measurements of ammonia ($NH_3$) along with water-soluble inorganic ions in $PM_{2.5}$ ($Cl^-$, $NO_3^-$, $SO_4^{2-}$ and $NH_4^+$) and corresponding precursor gases (HCl, $SO_2$, HONO, and $HNO_3$) made at the WiFEX research site, using the Monitor for AeRosols and Gases in Ambient Air (MARGA) and high-resolution simulations with Weather Research and Forecasting model coupled with chemistry (WRF-Chem). The hourly measurements were used to investigate how well the model captures the temporal variation of gaseous and particulate water-soluble species and gas-to-particle partitioning of $NH_3$, using the Model for Simulating Aerosol Interactions and Chemistry (MOSAIC) aerosol scheme. The model frequently simulated higher $NH_3$ and lower $NH_4^+$ concentrations than the observations, while total $NH_x$ values/variability agreed well with the observations. Under the winter conditions of Delhi, high concentrations of hydrochloric acid (HCl) in the ambient air are found to dominate the gas-to-particle partitioning, as $NH_3$ is usually in excess. The default model set-up of WRF-Chem excludes anthropogenic HCl emissions, so sulfuric acid ($H_2SO_4$) dominates the gas-to-particle partitioning with $NH_3$ during the simulation period. The sensitivity experiments, including HCl emissions in the model, showed that the inclusion of HCl emissions improves the simulated gas-to-particle conversion rate of ammonia by 24 % (as indicated by $NH_4^+$ concentrations) while reducing the bias in gas phase $NH_3$ by 10 %. Nevertheless, even with waste burning HCl emissions included, we find that WRF-Chem still overestimates sulfur dioxide ($SO_2$) and nitrate ($NO_3^-$) formation and underestimates sulfate ($SO_4^{2-}$), nitrous acid (HONO), nitric acid ($HNO_3$), and HCl concentration in which it interacts, thus limit the gas-to-particle conversion of $NH_3$ to $NH_4^+$ in the model. This indicates that modeling of ammonia requires a correct chemistry mechanism with accurate emission inventories for the industrial HCl emissions.



## 1 Introduction

The Indo-Gangetic Plain (IGP) has been one of the global hotspots of atmospheric ammonia (NH$_3$) and faces a range of environmental challenges, particularly during the winter season, including adverse air pollution episodes, as NH$_3$ plays a substantial role in secondary aerosol formation (Ghude et al., 2020, 2008b, 2008a; Kumar et al., 2021; Saraswati et al., 2019; Sharma et al., 2020; Singh et al., 2021). Atmospheric NH$_3$ along with oxides of nitrogen (NO$_x$) together account for the largest source of reactive nitrogen ($N_r$), which is primarily emitted by agricultural activities, livestock population, industrial activities, and transportation (Ghude et al., 2009, 2010, 2012, 2013; Móring et al., 2021; Pawar et al., 2021; Sutton et al., 2017b). NH$_3$ in the environment plays a crucial role in atmospheric chemistry and the eutrophication and acidification of ecosystems (Datta et al., 2012; Mandal et al., 2013; Pawar et al., 2021; Sharma et al., 2008, 2012, 2014b). Control of ammonia becomes a key priority in an emerging international strategy to manage the global nitrogen cycle (Gu et al., 2021; Sutton et al., 2020). Ammonia is a significant precursor of an aerodynamic diameter smaller than 2.5 μm (PM$_{2.5}$) containing ammonium sulfate, ammonium nitrate, and in some environments, ammonium chloride (Seinfeld and Pandis, 2016). In addition, as the dominant alkaline gas in the atmosphere, NH$_3$ has attracted the interest of scientific researchers since it has been known to promote new aerosol formation both in the initial homogeneous nucleation and in the subsequent growth, especially during wintertime (Acharja et al., 2020, 2021; Ali et al., 2019; Duan et al., 2021; Wagh et al., 2021).

In this study, we focus on wintertime analyses since this season is characterized by low-to-dense fog events, lower temperature (T), and variability of relative humidity (RH), which fluctuates from 40 to 100 % (Ghude et al., 2017; Kumar et al., 2020). Ammonia acts as a neutralization agent for determining the acidity of aerosol particles (Acharja et al., 2020; Ali et al., 2019; Ghude et al., 2017). It also affects the PM$_{2.5}$, acidity of clouds, and wet deposition of nitrogen by neutralizing acidic species (Gu et al., 2021; Xu et al., 2020). Increasing NH$_3$ concentration over Delhi compared with the surrounding area leads to an increase in PM$_{2.5}$ concentrations (Sharma et al., 2008, 2012, 2014a), which in turn affects air quality, human health, and climate (Behera et al., 2013; Ghude, 2016; Ghude et al., 2008b; Nivdange et al., 2022; Sutton et al., 2017a; Sutton and Howard, 2018).

Satellite observations (Van Damme et al., 2018; Warner et al., 2017), chemical transport models (Clarisse et al., 2009, 2010; Wang et al., 2020b), and ground-based observations (Pawar et al., 2021) revealed that the IGP is the largest regional hotspot of NH$_3$ concentrations on the Earth. Previous studies have identified various sources of NH$_3$ such as agricultural activities, industrial sectors, motor vehicles, garbage, sewage, and urine from rural populations, etc. at the global scale (Behera et al., 2013; Huang et al., 2012; Sutton et al., 2008). However, in Delhi, agricultural activity (including surrounding arable and sub-urban livestock farming) is the dominant source of NH$_3$ along with the traffic emissions(Kuttippurath et al., 2020; Móring et al., 2021; Sharma et al., 2020) and its emissions are subject to large uncertainty. Globally, various modeling efforts have investigated the relative effectiveness of reducing NH$_3$ emissions in curtailing PM$_{2.5}$ formation (Gu et al., 2021; Pinder et al., 2007, 2008; Zhang et al., 2020). However, over India, the impact on reducing PM$_{2.5}$ might be limited because NH$_3$ emission reductions may be more challenging due to its alkaline nature and area-wide sources. Ianniello et al. (2010) and Lan et al. (2021) have investigated the variation of atmospheric ammonia at an urban and suburban site of Beijing with respect to meteorological factors, where RH was found to be a strong factor for influencing the NH$_3$ mixing ratio. A few studies over Asia have highlighted the gas-to-particle



conversion of $NH_3$ in Delhi (Acharja et al., 2021; Saraswati et al., 2019) and China and its subsequent impact on
aerosol formation (Wang et al., 2015; Xu et al., 2020). Furthermore, excess ammonia during fog can also
enhance secondary aerosol formation in Delhi during winter (Acharja et al., 2021). However, the wintertime
behavior of ammonia in Delhi in chemical transport models (CTM) has not yet been investigated and remains
poorly understood (Ellis et al., 2011; Metzger et al., 2006). In a recent study, Pawar et al. (2021) has highlighted
uncertainties associated with gas-to-particle partitioning of $NH_3$ in a global model MOZART-4 and found a
significant overestimation of $NH_3$ in the model compared with the measurements. The overestimation of
ammonia in the model led the authors to hypothesize that a source-specific ammonia emission inventory in India
considering agricultural statistics on the fertilizer use and animal distribution was missing. Also, there was a
need for a high-resolution regional model with advanced chemistry to resolve the $NH_3$ emissions on the local
scale.
The present study utilizes measurements from the Winter fog Experiment (WiFEX), including $NH_3$,
water-soluble ions in $PM_{2.5}$, other trace gases, and meteorological parameters, interpreted using the regional
Weather Research and Forecasting model coupled with chemistry (WRF-Chem) during December-January
2017-18. For the first time in India, we discuss and compare the observed and modeled temporal variation in
$NH_3$, $NH_4^+$, and total ammonia ($NH_x$). Since we found that modeled $NH_x$ matches well with the observations,
we investigate the measurements and modeling associated with the gas-phase $NH_3$ and particulate $NH_4^+$ in terms
of gas-to-particle partitioning. We carried out several sensitivity experiments with and without the addition of
anthropogenic waste burning emissions of hydrochloric acid (HCl) in the model. The updated model with HCl
emissions was used to analyze and compare the temporal variation of $NH_3$, $NH_4^+$, and $NH_x$ from the WiFEX
measurements.
**2. Data and methodology**
**2.1 Observational datasets**
In this study, we used Monitor for AeRosols and Gases in Ambient Air-model 2S (MARGA), having
two channels, one for sampling $PM_1$ and the other channel sampling $PM_{2.5}$ for the ground-based observations.
The air was first passed through two dedicated separate impactors (cut-off diameters 1 and 2.5 μm respectively)
with the exit air (as $PM_1$ and $PM_{2.5}$) then sent through 1 cm long diameter PolyTetraFluoroEthylene (PTFE)
tubes to the MARGA to achieve the separation of $PM_1$ and $PM_{2.5}$. The flow rate in each sampling box is
regulated to a volumetric flow of 1 $m^3$ $h^{-1}$. Anions are separated in a Metrosep A Supp 10 (75/4.0) column,
whereas for cations separation, a Metrosep C4 (100/4.0) cation column is used (Acharja et al., 2020). The
MARGA was located inside the Indira Gandhi International Airport (IGIA), New Delhi (28.56° N, 77.09° E),
with 1 cm long inlet lines sampling outdoor air at 8 m above ground and 2 m above the rooftop. Measurements
covered a winter period (19[th] December 2017 to 21[st] January 2018) with frequent moderate to dense fog events.
Details of the MARGA instrument can be found in Makkonen et al. (2012), Thomas et al.(2009) and Twigg et
al. (2015).
Surface measurements of ambient ammonia were made along with other trace gases (HCl, HONO,
$HNO_3$, and $SO_2$) and water-soluble inorganic components ($Cl^-$, $NO_3^-$, $SO_4^{2-}$, $NH_4^+$, $Na^+$, $K^+$, $Mg^{2+}$, $Ca^{2+}$) of $PM_1$
and $PM_{2.5}$ at an hourly resolution. In this study, we have focussed on only $PM_{2.5}$ inorganic water-soluble



components for consistency with the selected WRF-Chem aerosol size distribution. The collected samples were
analyzed in the analyzer box using the ion-chromatography (IC) technique to analyze the chemical species. For
detailed information on the measurement site and its meteorological parameters, refer to (Ali et al., 2019). All
the precautionary measures were taken to minimize the manual errors in preparing the internal standard solution,
absorbing solution and regenerant, anion, and cation eluents to ensure the quality control of the data obtained
from MARGA. The $PM_{2.5}$ impactors were typically cleaned fortnightly to remove any material that may stick on
the surface and inlets of the impactors. The quality of the data obtained was then checked using the ion-balance
method. Further detail on the quality control of MARGA can be found in Acharja et al. (2020).
Hourly $NO_x$ measurements were made by the chemiluminescence method, and hourly ozone ($O_3$)
measurements were made by the UV photometric method (CPCB, 2011) at the nearest air quality monitoring
station (AQMS) of IGIA operated by the Central Pollution Control Board (CPCB). $NO_x$ analyzers contain a
thermal converter that catalytically reduces $NO_2$ to NO. The original NO and the NO (converted from $NO_2$) in
the sample are then reacted with ozone ($O_3$) to give a total $NO + NO_2$ ($NO_x$) reading. At the same time, an ozone
photometric analyzer measures $O_3$. These air quality monitoring stations' quality control and assurance
processes were followed as outlined in CPCB (2014, 2020). For data quality, we rejected all those observed
values which fell below the lowest detection limit of the instrument (1 µg m$^{-3}$ for $NO_x$ and 4 µg m$^{-3}$ for $O_3$)
(Technical specifications for CAAQM station, 2019) and above 500 µg m$^{-3}$ for $NO_x$ and 140 µg m$^{-3}$ for $O_3$ at a
given site. This step aims to remove any short-term local influence that cannot be captured in the models and
retain the regional-scale variability because the nearest sites are located in the urban environment. We removed
single spike represented by a change of more than 100 µg m$^{-3}$ in just 1 hour (h) for all the data in CPCB
monitoring stations to filter out random fluctuations in the observations. We removed some very high $NO_x$ and
$O_3$ values that appeared in the time series right after the missing values. Meteorological parameters, including
air temperature (T), relative humidity (RH), wind speed and wind direction were measured with the automatic
weather station (AWS) platform on a 20 m flux tower tower (Ghude et al., 2017).
**2.2 WRF-Chem v 3.9.1 model**
The Weather Research and Forecasting model coupled with chemistry (WRF-Chem v3.9.1) has been employed
in this study to simulate atmospheric gases and aerosols over Delhi during the peak winter period, starting from
19 December 2017 to 21 January 2018. We recently used a similar model configuration to simulate the air
quality over Delhi (Ghude et al., 2020; Kulkarni et al., 2020). This study used the MOZART-4 gas-phase
chemical mechanism coupled with the Model for Simulating Aerosol Interactions and Chemistry (MOSAIC)
aerosol scheme, including sulfate, ammonium, nitrate, methanesulfonate, sodium, calcium, chloride, carbonate,
black carbon, primary and organic mass. Other inert minerals, trace elements, and inorganic species are lumped
together as different inorganic masses. MOSAIC allows gas-to-particle formation, which includes $NH_3$, HCl,
sulfuric acid ($H_2SO_4$), nitric acid ($HNO_3$), and methane sulfonic acid (MSA), and also includes secondary
organic aerosols (SOA). Aerosol size distributions are represented by a sectional aerosol bin approach with four
size bins (Georgiou et al., 2018). MOSAIC includes a thermodynamic module named 'Multicomponent Taylor
Expansion Method' (MTEM), which is used for the calculation of the activity coefficients in aqueous
atmospheric aerosols, and the 'Multicomponent Equilibrium Solver for Aerosols' (MESA), which provides a
computationally efficient solution of the intraparticle solid-liquid phase equilibrium. This combined





thermodynamic module MESA-MTEM is coupled with the new gas-to-particle partitioning module named
'Adaptive Step Time-split Euler Method' (ASTEM), which dynamically integrates the mass transfer equations.
The ASTEM algorithm produces smooth, accurate and computational efficient solutions by solving the dynamic
gas-to-particle conversion at low, moderate and high RHs when aerosol particles may be entirely mixed or solid
phase. A new concept of "dynamic pH" along with an adaptive time stepping scheme in ASTEM algorithm
reduces the stiffness in the Ordinary Differential Equations (ODEs) and produces the explicit solutions over the
entire RH range. However, secondary organic aerosol (SOA) formation and their interactions with former
inorganic aerosol species, primary organics, water, and particle pH are not included in MOSAIC. Moreover,
their impact on gas-to-particle partitioning is poorly understood and remains an area of enormous uncertainty in
chemical transport models. For further details on the MOSAIC scheme, please refer to Zaveri et al. (2008).
The model domain covers the entire northern region of India, but here model simulations are compared
with the observations at IGIA, New Delhi (28.56° N, 77.09° E). The domain was set with a horizontal grid-
spacing of 10 km in both the latitudinal and longitudinal directions. The model top included 47 vertical levels
and was set to 10 hPa. The physical parameterization schemes of model configuration are the same as described
in Ghude et al. (2020) and Jena et al. (2021). EDGAR-HTAP (Emission Database for Global Atmospheric
Research for Hemispheric Transport of Air Pollution) for the year 2010 at 0.1° x 0.1° grid resolution has been
used in this study for anthropogenic emissions of aerosols and trace gases ($PM_{2.5}$, $PM_{10}$, OC, BC, CO, $NO_x$, etc.)
and are scaled to 2018 as per Jena et al. (2021). Biogenic emissions are calculated online using the Model of
Emissions of Gases and Aerosols from Nature version 2.1 (MEGAN2.1) (Guenther et al., 2006), and dust
emissions are based on online Atmospheric and Environmental Research Inc. and Air Force Weather Agency
(AER/AFWA) scheme (Ginoux et al., 2001). Fire INventory from NCAR (FINNv1.5) has been used in this
study for daily open biomass burning emissions. The chemical initial and lateral boundary conditions come from
the global model simulations from the Model for Ozone and Related Chemical Tracers (MOZART-4), and the
meteorological initial and lateral boundary conditions are provided from the fifth generation European Centre
for Medium-Range Weather Forecasts (ECMWF) atmospheric reanalysis of the global climate (ERA5) with six-
hourly temporal resolution. The simulations were reinitialized every fifth day to limit the growth of
meteorological errors in our simulations, but the chemical fields were carried forward from the previous
simulation.
**3. Results and Discussion**
**3.1 MARGA**
**3.1.1 Temporal variation in $NH_3$ and $NH_4^+$**
Figure 1 displays the diurnal variation (00:00 to 23:00 Indian Standard Time (IST)) in $NH_3$ and $NH_4^+$ averaged
over all the study period (Fig. 1a) along with meteorological parameters (temperature and RH) at the IGIA site
in Delhi (Fig. 1b). As indicated in Fig. 1a, the average $NH_3$ concentration maxima and minima were observed
during 08:00-12:00 h and 01:00-07:00 h, respectively. The mean ± 1σ, median, maximum and minimum values
in the average diurnal $NH_3$ concentration for the observation period were 28.20 ± 12.37, 28.45, 40.98, and 20.09
μg m$^{-3}$ respectively. The $NH_3$ concentration gradually increased during 17:00-1:00, decreased from 1:00-7:00 h,





194 and then rapidly increased from 08:00 h (just after sunrise). After reaching the peak at approximately 12:00 h, a

195 decrease was observed until it came to the minimum of 25.65 µg m$^{-3}$ at 15:00 h. The mean ± 1σ, median,

196 maximum and minimum values of the average diurnal $NH_4^+$ concentration during the study period were 36.96 ±

197 15.10, 39.25, 46.28, and 21.45 µg m$^{-3}$ respectively. While the average $NH_4^+$ concentration maxima and minima

198 were observed during night time (16:00-03:00 h) and daytime (03:00-08:00 and 09:00-16:00 h), respectively.

199 The daytime increase in $NH_3$ concentration could be associated with $NH_4^+$ aerosol volatilization driven by

200 associated increases in temperature, which was observed mainly from 08:00 h onwards. This indicates that an

201 increase in $NH_3$ concentration in the morning is caused by meteorological conditions. From Fig. 1b, RH (T) was

202 observed to be relatively constant before 08:00 h but decreased (increased) sharply in the later morning. High

203 temperature and low RH contribute significantly to the evaporation of $NH_3$ from ammonium volatilization

204 (Acharja et al., 2020; Sutton et al., 2009, 2013), indicating gas-to-particle partitioning impacts the diurnal

205 behavior of ammonia at Delhi during winter. Recent studies have confirmed that evaporation of $NH_3$ from

206 plants, human sources, plant stomata, morning traffic, fertilized soil, mixing down of ammonia concentration

207 from the residual layer and dew contributes to the morning increase in $NH_3$ (Ellis et al., 2011; Meng et al., 2018;

208 Norman et al., 2009; Sutton et al., 2001).

209  The average diurnal profile of $NO_x$ from the nearby CPCB site (RK Puram station) is displayed in Fig.

210 S1 in the Supplement. A similar variation for $NO_x$ was also observed by Chate et al. (2014) and Ghude et al.

211 (2008a) over Delhi in the previous studies. It can be seen that $NO_x$ concentrations increase in the morning till

212 9:00 h, followed by a sharp decrease in the afternoon, and reach maximum concentrations in the evening around

213 17:00-19:00 h. Morning and evening peak coincides with the rush hour of traffic flow during the peak hour.

214 However, it is evident that the observed morning peak of $NH_3$ was not concurring with corresponding $NO_x$

215 peaks, suggesting that traffic emissions do not contribute significantly to the observed $NH_3$ rise.

216  During dense fog events, a significant amount of dew formation and deposition occurs on the surface is

217 common during winter over Delhi (Ghude et al., 2017). Studies have shown that a significant amount of

218 ammonia trapped in liquid water condensed on surfaces is released due to evaporation of dewdrops after sunrise

219 Sutton et al. (1998). Hence, we examine how extreme dense fog and non-fog events (clear days) impact the

220 diurnal variation of $NH_3$. Figure S2 in the Supplement shows the diurnal variation of $NH_3$ on dense fog events

221 (13 days), which sustain more than four hours with visibility less than 200 m and on clear days with visibility

222 more than 1000 m (20 days). Figure S2 indicates that the $NH_3$ morning spike phenomenon is most prevalent

223 during the extreme dense fog events compared to clear days. The magnitude of increase in $NH_3$ during foggy

224 days is two times higher with a maximum of 65 µg m$^{-3}$ compared to the clear days with maxima 35 µg m$^{-3}$,

225 indicating strong evidence that dew evaporation and the early morning $NH_3$ increases are linked The observed

226 $NH_3$ morning peak phenomenon during foggy days suggests that dew droplets may also act as an adequate $NH_3$

227 night-time reservoir and emission sources during the day. However, we are not focussing on this aspect in the

228 manuscript. The observed increase in morning peak (08:00 h) at IGIA could be explained by the release of $NH_3$

229 from the dew evaporation. Moreover, the contribution of dew evaporation and guttation water droplets for

230 causing morning $NH_3$ peaks requires further investigation.

231  Since the IGIA site is surrounded by the local agricultural activities, including surrounding arable and

232 sub-urban livestock farming should be a prime source contributing to $NH_3$ concentration levels (Kuttippurath et

233 al., 2020) in addition to dew acting as a night-time reservoir of $NH_3$ and subsequently a morning source for





atmospheric ammonia. Hence, in this study, analysis from the above aspects such as ammonium volatilization
and dense fog events pointed to the conclusion that the ammonium volatilization and evaporation processes of
dew and fog water were the cause for the morning rise of ammonia. Potential sources of $NH_3$ and $NH_4^+$ are
discussed further in the next section.
**3.1.2 Potential sources of $NH_3$ and $NH_4^+$**
We investigated the directions of local emission sources associated with the rise in $NH_3$, $NH_4^+$ concentrations,
and partitioning ratio of $NH_4^+$ to a total ammonia ($NH_x = NH_3 + NH_4^+$) ($NH_4^+/NH_x$) by the bivariate polar
graphs using the OpenAir software (Carslaw and Ropkins, 2012) at the IGIA site. Figure 2 shows the wind rose
(Fig. 2a) and bivariate polar plots of mean $NH_3$ (Fig. 2b), $NH_4^+$ (Fig. 2c), hydrochloric acid (HCl) (Fig. 2d),
chloride ($Cl^-$) (Fig. 2e) concentration, and partitioning ratio of $NH_4^+$ to total ammonia ($NH_x = NH_3 + NH_4^+$)
($NH_4^+/NH_x$) (Fig. 2f ) for the observation period in relation to wind speed and wind direction. The frequency of
local wind indicates that the dominant direction from which air masses arrived at the IGIA site was from the
270-300º sector (Fig. 2a). The bivariate polar plot of gas-phase $NH_3$ in Fig. 2b shows that $NH_3$ emissions are
highest in the east and southeast of the site, indicating a dominant source of ammonia in that direction. The
IGIA site is surrounded by intense agricultural areas having various agricultural and fertilizer sources located
within 200 km of Delhi and towards the east and southeast of the site (Kuttippurath et al., 2020). Additionally,
large numbers of local dairy farms are also situated towards the east and southeast of the site. The discharge of
wastewater and cow dung into the drains from the local dairies can contribute a major portion of agricultural
ammonia emissions since 66 dairies were shut down recently by The Delhi Pollution Control Committee
(DPCC) for violating environmental norms and polluting water bodies in southeast Delhi (Hindustan Times,
2021). This enhancement in the southeast region may also be related to lower wind speed, falling in the leeward
side of the wind direction. Ammonia concentrations are generally higher at lower wind speeds because of
turbulent diffusion (Ianniello et al., 2010).
The bivariate polar plot of $NH_4^+$ (Fig. 2c), HCl (Fig. 2d), and $Cl^-$ (Fig. 2e) concentration indicates west
and north-west direction as a principal source from various industries and thermal power plants which are
located towards the same direction of IGIA site (Acharja et al., 2021; Kuttippurath et al., 2020; Singh et al.,
2021). The ratio of $NH_4^+/NH_x$ has been used to identify the source of $NH_x$ and the relative contribution of $NH_3$
to aerosol formation. A higher ratio indicates that $NH_x$ is primarily due to the conversion of $NH_3$ to salt has
proceeded and thus dominates the $NH_x$. The polar plot of the $NH_4^+/NH_x$ ratio in Fig. 2f identified sporadic high
ratio originating from the prevailing wind direction of the IGIA site in the west direction with lower wind speed
which is associated with the formation of ammonium chloride (Fig. 2c and e) from the local industrial source of
HCl located north-west (NW) of the site (Acharja et al., 2021). Two sources stand out-an industrial clusters in
NW Delhi, such as steel pickling industries which are known to be a vital HCl source, and metal finishing and
electroplating towards the west, which impacts the site (falling in the region of high wind speed) (Fig. 2d and e)
(Jaiprakash et al., 2017). Near the source, HCl is immediately available for neutralizing $NH_3$ (Fig. 2b and d),
hence high $NH_4^+/NH_x$ ratio towards the west (Fig. 2f) indicates that $NH_3$ neutralization by HCl is more critical,
which dominates ammonium chloride formation (Fig. 2c and e) at lower wind speeds. The nearest industrial
areas are located ~8–9 km Northwest of IGIA site houses industrial units related to metal product manufacture,
scrap metal processing, plastic, rubber, pigment industries, etc. Emissions from these industrial processes and/or




fuel oil combustion in these areas might be necessary at the study site. Thus, high $NH_4^+/NH_x$ correspond to the
lowest $NH_3$ concentration region (inverse relation), which can be observed in Fig. 2b and f and also correspond
to the highest chloride concentration region (Fig. 2b and e).

### 3.2 Comparison of temporal variation in $NH_3$, $NH_4^+$, and $NH_x$ using WRF-Chem and MARGA

#### 3.2.1 Diurnal variation

To investigate how well a state-of-the-art chemical transport model performs in capturing the diurnal behavior
of $NH_3$ and $NH_4^+$, we looked at the observed and model-simulated diurnal profiles of $NH_3$ and $NH_4^+$. We
adopted diurnal variation in emissions from a recent study by Jena et al. (2021). We first investigated the ability
of WRF-Chem to accurately predict the meteorological parameters of RH and T, which are important
determinants of the gas-to-aerosol partitioning of volatile compounds. As shown in Fig. 3b and 8 (k and l), the
simulated temperature and relative humidity are in reasonable agreement with the observations, with the
simulated RH values falling in the range of 50–90 %.
Figure 3a shows the comparison of diurnal variations in the $NH_3$ and $NH_4^+$ concentration
between MARGA measurements and model simulations for the observation period. Figure 3a shows that the
model simulated $NH_3$ and $NH_4^+$ are very different compared with the MARGA observations. There is an
average $NH_3$ and $NH_4^+$ mass loading of 56.75 ± 14.28 and 14.71 ± 4.90 µg m$^{-3}$ respectively, in the model. We
find the diurnal variation of gas-phase $NH_3$ is significantly overestimated by the model (Normalised Mean Bias
(NMB) = 1.02). On the contrary, $NH_4^+$ is underestimated by about 60 % (NMB = -0.60). Simulated ammonia
concentrations peak between 7:00-9:00 and 22:00-23:00 h with bimodal variation, though observations show a
single peak around 12:00-13:00 LT. On the contrary, a nearly flat diurnal profile of $NH_4^+$ is predicted by the
model. Figure 3b depicts that overall the model shows cold and wet bias compared to the observations but
shows warm bias (about 2-3 ℃) and dry bias (about 10-12 %) in the afternoon hours. In spite of the small
change of the amplitude of the diurnal cycle of RH, the phase characteristics of the diurnal cycle of both
temperature and relative humidity are reasonably well captured by the model. Because the largest increase in
simulated ammonia also precedes the large changes in simulated meteorological parameters, which is not the
case in the model, and because the simulated particulate ammonium is flat compared to observations, simulated
meteorology is ruled out as a significant contribution to high bias in simulated ammonia. It is hence necessary to
understand other processes responsible for the overestimation of $NH_3$ and underestimation of $NH_4^+$ in the model.

#### 3.2.2 Daily mean variations

Figure 4 displays the comparison of daily mean variations in the $NH_3$ (Fig. 4a), $NH_4^+$ (Fig. 4b), and $NH_x$ (Fig.
4c) concentration between MARGA measurements and model simulations. The model shows large differences
in $NH_3$ and $NH_4^+$ compared with MARGA (Fig. 4a and b). We find a consistent high positive bias in daily mean
$NH_3$ (NMB = 1.02 with fair correlation, $r = 0.54$) in the model and negative bias in $NH_4^+$ (NMB = -0.62) with
poor correlation ($r = -0.08$). Whereas, $NH_x$ (Fig. 4c) shows low bias (NMB = 0.08, $r = 0.60$) between the model
and observation. Despite the adequate ability of the model to reproduce the total amount of the measured
ammonia ($NH_x$), the model is biased low for $NH_4^+$ and high for $NH_3$, indicating that incorrect gas-to-particle
partitioning could be responsible for the large bias for $NH_3$ in the model. To understand if gas-to-particle



partitioning is the main cause for the discrepancy in the model vis-a-vis the observations, comparing the ratio of
gas-to-particle partitioning of ammonia in the measurements can help elucidate the contributing process.

### 3.3 Gas-to-particle partitioning of NH$_3$

Figure 5 shows the percentage contribution of gases (NH$_3$, SO$_2$, HCl, HNO$_3$, and HONO) and PM$_{2.5}$ aerosol
(NH$_4^+$, SO$_4^{2-}$, NO$_3^-$ and Cl$^-$) during WiFEX measurements. The pie charts for the gases show that a high amount
of NH$_3$ (53.28 %) is a significant gas, followed by sulfur dioxide (SO$_2$) (35.61 %). Ammonia is one of the
highly alkaline gases that act as a precursor for aerosol formation. In the atmosphere, NH$_3$ reacts rapidly with
H$_2$SO$_4$, HNO$_3$, and HCl to form ammonium sulfates ((NH$_4$)$_2$SO$_4$, and in low NH$_3$ environments also NH$_4$HSO$_4$),
ammonium nitrate (NH$_4$NO$_3$), and ammonium chloride (NH$_4$Cl), respectively. The principal reactions of NH$_3$
with H$_2$SO$_4$, HNO$_3$, and HCl in the gas-to-particle partitioning process to produce ammonium salts of PM$_{2.5}$ are
summarized in reactions R1 to R3 (Seinfeld et al., 1998).

$2NH_{3(g)} + H_2SO_{4(g)} \leftrightarrow (NH_4)_2SO_{4(s)}$                                          (R1)
$NH_{3(g)} + HNO_{3(g)} \leftrightarrow NH_4NO_{3(s)}$                                          (R2)
$NH_{3(g)} + HCl_{(g)} \leftrightarrow NH_4Cl_{(s)}$                                          (R3)

Figure 5 shows that HCl concentration (1.09 %) is comparatively low than SO$_2$ (35.61 %), but HCl is
immediately available for neutralizing NH$_3$. Hence near the sources (Fig. 2d), NH$_3$ neutralization by HCl (R3) is
more critical than H$_2$SO$_4$ neutralization (or indeed HNO$_3$). Also, the formation of NH$_4$Cl (R3) is favored under
conditions of high relative humidity and low temperature (Ianniello et al., 2011; Seinfeld and Pandis, 2016),
which was the case during WiFEX. Hence, as indicated in Fig. 5, on average, the anions in PM$_{2.5}$ are almost
exactly neutralized by NH$_4^+$ (49.54 %), with Cl$^-$ (29.69 %) as the primary anion. But the pie chart for the gases
shows a very high amount of SO$_2$, reaching the site from the nearby industrial area, which is not converting to
SO$_4^{2-}$ very quickly. In the atmosphere, gas-phase oxidation of SO$_2$ occurs either through its reaction with the
hydroxyl radical (OH) or in the aqueous phase, through reaction with hydrogen peroxide (H$_2$O$_2$) or ozone (O$_3$)
(Luhana et al., 2007). However, in an ordinary ammonia-rich atmosphere, SO$_2$ oxidation to SO$_4^{2-}$ due to O$_3$
reaction is faster than the typical gas-phase reaction rates at high pH (pH >= 4) (Seinfeld and Pandis, 2016) and
could contribute ~ 51 % to the total sulfate production (Li et al., 2020), while during fog days, the two
mechanisms have been found to dominate SO$_2$ oxidation such as the S(IV) to S(VI) conversion catalyzed by
NO$_2$ and HONO (Yang et al., 2019) and the S(IV) oxidation catalyzed by transition metal ions (TMI) (Harris et
al., 2013) which could contribute to the rest 49 % in the sulfate production (Li et al., 2020), but their
quantification over our study location needs to be carried out in the future (Wang et al., 2020a).

342         The primary SO$_2$ sources are located NW of the site (Acharja et al., 2021), which limits SO$_2$ oxidation

to SO$_4^{2-}$ in particular for fresh plumes originating in a nearby industrial area (Jaiprakash et al., 2017).
Additionally, during this study period, average O$_3$ concentrations (46.78 µg m$^{-3}$) are also low daytime (10:00-
17:00 LT) at the IGIA site (Fig. S3 in the Supplement) compared to the nearest CPCB site (86.65 µg m$^{-3}$); thus,
SO$_2$ oxidation to SO$_4^{2-}$ is limited during WiFEX.





To gain insight into the role of $NH_4^+$ in the neutralization of anions ($Cl^-$, $NO_3^-$ and $SO_4^{2-}$), aerosol
neutralization ratio (ANR) was calculated using the observed data. The ANR is defined as the equivalent ratio of
$NH_4^+$ to the sum of $Cl^-$, $NO_3^-$ and $SO_4^{2-}$ because these species represent the dominant cations and anions in
$PM_{2.5}$, respectively. We consider these four significant ions since they constituted 97.3 % of the total measured
ions in $PM_{2.5}$ and the remaining ionic species (i.e., $Na^+$, $K^+$, $Mg^{2+}$, and $Ca^{2+}$) contributed only about 3 % of the
total measured ions (Acharja et al., 2020). Figure 6 demonstrates how well the charge balance between $Cl^-$, $NO_3^-$
and $SO_4^{2-}$ (in $\mu eq\ m^{-3}$) as the anions and ANR works. It can be seen that acidic components were utterly
neutralized by $NH_4^+$ with ANR values close to unity with $Cl^-$ as a significant anion followed by $NO_3^-$ and $SO_4^{2-}$.
The average ANR value for $PM_{2.5}$ during the observed period was $1.08 \pm 0.16$. This indicates the winter period
was favoring the dominant formation of hygroscopic ammonium salts ($NH_4Cl$, $NH_4NO_3$, and $(NH_4)_2SO_4$).
The ratio of $NH_4^+/NH_x$ is calculated to evaluate the measured and modeled conversion rate of ammonia
to ammonium formation. Previous studies have reported the role of $NH_3$ in the $NH_4^+/NH_x$ ratio (Pawar et al.,
2021; Saraswati et al., 2019; Wang et al., 2015). Here, we compare MARGA measurements and model results at
the IGIA site to investigate the conversion rate of ammonia to aerosol formation for the study period. Figure 7
shows the relationship between measured (Fig. 7a and b)  and modeled (Fig. 7c and d) gas-to-particle
conversion ratio ($NH_4^+/NH_x$) with dominant cation ($NH_4^+$) and anion ( $Cl^-$) (Fig. 7a and b) and dominant cation
($NH_4^+$) and anion ($SO_4^-$) mass concentration (Fig. 7c and d). In Fig. 7a, $NH_4^+/NH_x$ was found to be directly
proportional to $NH_4^+$ concentration with correlation coefficient ($r$) = 0.79, and in Fig. 7b, compared to other
aerosols (statistical indicators are summarised in Table 1 (first column)), $NH_4^+/NH_x$ was found to be directly
proportional to chloride concentration ($r$ = 0.79) which were significant at IGIA site for the observational period
(Fig. 5 and 6) (Acharja et al., 2021). This indicates high conversion rate of ammonium from gaseous to particle-
phase enhanced the formation of chloride aerosol during the WiFEX. The strong correlation coefficient of
ammonium and chloride concentration explains that $PM_{2.5}$ is composed predominantly of converted ammonium
and chloride from the reactions of ammonia with HCl (Fig. 5). Furthermore, $NH_4^+/NH_x$ was inversely
proportional to the ambient ammonia ($r$ = -0.57) and nitrous acid (HONO ) ($r$ = -0.24) concentrations (Table 1
(first column)), indicating atmospheric inter-conversion of $NH_x$ between its gas and particle phases. The co-
variability of $NH_3$ and HONO suggests that the abundance of $NH_3$ may enhance HONO formation, which in
turn could increase secondary aerosol formation (Fu et al., 2019; Ge et al., 2019). However, in a future study
understanding the role of HONO in promoting secondary aerosols via the heterogeneous reactions pathway is
required.
The relationship between the $NH_4^+/NH_x$ with all gases and aerosols is investigated to understand the
role of gas-to-particle conversion of $NH_3$ in the model, and statistical values are summarized in Table 1 (second
column). The role of ambient $NH_3$ in the formation of $(NH_4)_2SO_4$ is indicated by the significant positive linear
relationship of $NH_4^+/NH_x$ ratio with ammonium ($r$ = 0.67) and sulfate ($r$ = 0.77) in the model. Figures 7c and d
show the relationship between modeled $NH_4^+/NH_x$ ratio with ammonium and sulfate concentration. The high
conversion rate of ammonium from gaseous to particle-phase indicates the formation of sulfate, and then nitrate
($r$ = 0.57) aerosols are enhanced in the model (Table 2 (second column)). Since the MOSAIC scheme is based
on the order of increasing volatility of the associated ammonium compounds (Zaveri et al., 2008), hence $H_2SO_4$
(non-volatile) has a stronger affinity with $NH_3$, ammonia first reacts with it to form $(NH_4)_2SO_4$, then it reacts
with semivolatile gases $HNO_3$ to form $NH_4NO_3$ (R1 and R3). Hence, $NH_4SO_4$ was more likely to form first in





the model, as indicated by the higher correlation value of $NH_4^+/NH_x$ with ammonium and sulfate as compared to
$NH_4NO_3$. However, the observations at IGIA indicate that the presence of HCl concentration (Fig. 2d) enhances
chloride concentration (Fig. 2e) which plays a significant role in the gas-to-aerosol partitioning of $NH_3$ in Delhi
compared to other aerosols. Hence in order to improve the efficiency of the model in determining the gas-to-
particle conversion of $NH_3$ and subsequently improving $PM_{2.5}$ predictions, it is necessary to add anthropogenic
HCl emissions, which are currently missing in the default set-up of the model.
**3.4 Sensitivity case study on the effects of addition of HCl emissions on the simulated $NH_3$ in WRF-Chem**
We turn our attention to exploring the effects of the addition of anthropogenic HCl emissions on the simulated
$NH_3$ concentration. Other recent studies have identified significant concentrations of aerosol $Cl^-$ in cities across
India, and evidence suggests that this primarily derives from HCl emitted from the open trash burning (Cash et
al., 2021; Gani et al., 2019; Gunthe et al., 2021; Reyes-Villegas et al., 2021) and the nearby industrial sources
(Jaiprakash et al., 2017). We, therefore, employ the HCl emissions from trash burning activities in Delhi as
predicted by Sharma et al. (2019) in our model set-up. We have carried out a few sensitivity experiments for the
period $7^{th}$ to $16^{th}$ January 2018 (10 days) by including the HCl emissions in the model to study their impact on
the model simulations. We carried out three experiments for this purpose. In sensitivity experiment-1, we run
the model with default set-up without HCl emissions. In sensitivity experiment-2, we tripled the original HCl
emissions of Sharma et al. (2019) because the revised emission inventory was 2.9 times higher compared to
those in the Sharma et al. 2019 emission inventory. Hence, we consider this a base case of HCl emissions,
presuming the more recent upward adjustments in the amount of waste burned in landfills (Chaudhary et al.,
2021). In the third experiment (sensitivity experiment-3), we tripled the base case HCl emissions to take into
account missing industrial HCl sources. Figure 8 presents the box-whiskers plots for secondary inorganic
aerosols and trace gases from the MARGA, and meteorological parameters from the AWS and those simulated
by the model for the three different sensitivity experiments. As can be observed from Fig. 8a-c with increasing
HCl emissions, $NH_4^+$ concentrations increase, and $NH_3$ gas-phase concentrations decrease, chloride
concentrations increase drastically in all the three sensitivity experiments (Fig. 8f), and total $NH_x$ concentrations
increase slightly. Higher HCl concentrations promote the gas-to-particle conversion of excess $NH_3$, which in
turn enhances ammonium and chloride concentration in the sensitivity experiment-3. This is further discussed in
the next section. Increasing HCl emissions by a factor of three in the sensitivity experiment-3, the model
simulates $NH_4^+$ concentration (up to 70 $\mu g\ m^{-3}$) and $Cl^-$ concentration (up to 110 $\mu g\ m^{-3}$) that compare reasonably
well within the observed range, but continues to overestimate $NH_3$ concentration.

417         The simulated sulfate concentration (Fig. 8e) was underestimated mainly ($\sim 40 – 50 \%$) by the model

in all the experiments compared with the observations. The gas-phase $SO_2$ simulated by all the three model
experiments was found to be slightly overestimated by about 10 to 15 $\mu g\ m^{-3}$. This is caused by the fact that
neither the S(IV) to S(VI) conversion catalyzed by $NO_2$ nor (Yang et al., 2019) the S(IV) oxidation catalyzed by
TMI (Harris et al., 2013) is currently included in the MOSAIC mechanism. Wang et al. (2020) proposed a
mechanism in which dissolved $NO_2$ oxidizes S(IV) to S(VI) and produces HONO, which can either partition
into the gas phase or oxidize another molecule of S(IV) to S(VI). The latter reaction produces $N_2O$, which
partitions into the gas phase. The inclusion of this mechanism into the MOSAIC scheme is likely to significantly
improve the overestimation of $NH_3$, $SO_2$, and $NO_3^-$ as well as the underestimation of $SO_4^{2-}$ and HONO in the





model. Furthermore, sulfate is produced ($SO_2$ oxidized to sulfuric acid $H_2SO_4$) via aqueous-phase oxidation of
S(IV) by $O_3$, and a heterogeneous nucleation rate from sulfuric acid ($H_2SO_4$) is likely not efficiently simulated
by the model since ozone concentration is low in the model compare with the observations (Fig. S4 in the
Supplement). However, we also compared the cloud fraction of the model and from the Moderate Resolution
Imaging Spectroradiometer (MODIS) satellite, and we found that the lower cloud fractions (Fig. S5 in the
Supplement) could also lead to relatively weaker aqueous phase sulfate production and hence lower sulfate mass
in the model (Bucaram and Bowman, 2021; Sha et al., 2019).
The simulated nitrate concentration (Fig. 8d) is generally higher in all the three sensitivity experiments;
since the main neutralizing species for $NO_3^-$ is $NH_4^+$, it is controlled via the equilibrium between nitrate, $HNO_3$,
and $NH_3$. Simulated $HNO_3$ and HONO are significantly underestimated ($\sim$ 3-5 times) by the model compared to
the observations, due to which nitrate formation is entirely ruled by gaseous $NH_3$ and $HNO_3$. Also,
overestimating the $NO_x$ concentration in the model compared with the observations (Fig. S6 in the Supplement)
and incorrect gas-to-aerosol partitioning can allocate too much $NO_3^-$ into the aerosol phase, so it might not leave
enough in the gas phase. This could partly be a consequence of $NH_3$ being too high in the model and may
somewhat lead to these positive discrepancies (Bucaram and Bowman, 2021; Sha et al., 2019). The extent to
which sulfate is neutralized and $NH_4NO_3$ and $NH_4Cl$ is formed are mainly governed by thermodynamic
equilibrium, which is solved by MTEM-MSEA (Multicomponent Taylor Expansion Method and
Multicomponent Equilibrium Solver for Aerosols) module used in MOSAIC (Zaveri et al., 2008). Hence, excess
ammonia, after neutralizing $H_2SO_4$, consumes $HNO_3$ forming $NH_4NO_3$, which may further underestimate $HNO_3$
concentration.
Overall the missing heterogeneous chemistry in the formation of $HNO_3$ and HONO along with its gas-
to-particle partitioning to ammonium nitrate, may not be efficient in the model (Archer-Nicholls et al., 2014).
Hence, different treatments of the gas-to-particle partitioning from the nitric acid to ammonium nitrate as a
function of humidity (Balzarini et al., 2015; Georgiou et al., 2018) could result in biases in nitrate concentration
in the presence of higher $NH_3$ levels. In the future, an updated WRF-Chem model with the currently known
sources and chemistry of HONO and $HNO_3$ and its parameterization is required to accurately determine the
chemistry of $NH_3$, $NO_x$, and other aerosols (Zhang et al., 2017).
To investigate the further impact of sensitivity experiment-3 in the model, uptake of gaseous $NH_3$ to
form $NH_4Cl$ is analyzed by the correlation coefficient values of $r$ = 0.86 between $NH_4^+/NH_x$ with ammonium
concentration and $r$ = 0.84 between $NH_4^+/NH_x$ with chloride concentration, indicating gas-to-particle conversion
in the model correlates well with both the ammonium and chloride concentration. A higher ratio indicates that
the model is able to form $NH_4Cl$ from the conversion of $NH_3$ with HCl and was reasonably well simulated in the
sensitivity experiment-3. It can be noted that it required three times increased base case HCl emissions to be
able to form $NH_4Cl$ in the sensitivity experiment-3.
**3.5 Comparison of the temporal behavior of $NH_3$, $NH_4^+$, and $NH_x$ of MARGA with all the sensitivity**
**experiments of the model**
We further compare the diurnal behavior of $NH_3$, $NH_4^+$, and $NH_x$ using both the model and observations to
develop and improve model accuracy in simulating complex short-lived ammonia. Figure 9 presents the diurnal
variation of mean ammonia (Fig. 9a), ammonium (Fig. 9b), and total ammonia (Fig. 9c) concentration for all
three different sensitivity experiments. While the simulated $NH_3$ concentration decreases in sensitivity





experiment-3 compared to sensitivity experiment-1 concerning MARGA observations, none of the model
experiments capture the diurnal cycle of $NH_3$. Higher levels of observed $NH_3$ during daytime and modeled $NH_3$
during night-time highlight the need to develop diurnal variability in $NH_3$ emissions over this region. The
increased HCl emissions in sensitivity experiment-3 improved the linear correlation between modelled and
measured $NH_4^+$ concentration ($r = 0.76$) compared with the sensitivity experiment-1 ($r = 0.45$). By contrast, the
$NH_x$ (Fig. 9c) concentration does not significantly change its correlation coefficient between MARGA and
model with values of 0.69, 0.70, and 0.70 for the sensitivity experiments 1, 2, and 3, respectively. The NMB for
$NH_3$ reduced from 138 % to 113 %, NMB for $NH_4^+$ systematically improved from -61 % to 3 %. In contrast,
NMB for $NH_x$ increased from 12 % to 39 % for the simulations demonstrated by sensitivity experiments 1 and
3, respectively, in the model. Table 2 summarizes the statistical indicators for the three sensitivity experiments.
$NH_4Cl$ formation in the sensitivity experiment-3 leads to a higher mass concentration of ammonium, which also
increases $NH_x$ mass concentration by 27 %. We find consistent high bias in all the simulations of $NH_3$, which is
the highest during early morning and night time (Fig. S7 in the Supplement).
Figure S8 in the Supplement illustrates a time-series graph that compares daily mean ammonia (Fig. S8a),
ammonium (Fig. S8b), and total ammonia concentration (Fig. S8c) for the three different sensitivity
experiments. Table 3 shows the mean ± standard deviation of these variables. The results show that compared to
experiment 1, $NH_3$ concentrations decreased by 4 % in the sensitivity experiment-2 and further decreased by 10
% in the sensitivity experiment-3. On the contrary, $NH_4^+$ concentration improves in sensitivity experiment-2 by
54 % and further increases by 150 % (sensitivity experiment-3). This decrease in $NH_3$ is associated with the gas-
to-particle conversion of $NH_3$ to $NH_4^+$. Since $NH_4^+$ concentration seems to be very sensitive to the increase in
HCl emissions in the sensitivity experiment-3 (150 % improvement in $NH_4^+$), total ammonia also increased by
8.68 % and 24 % in the sensitivity experiment-2 and 3 respectively compared to the sensitivity experiment-1.
**3.6 Observed and modeled (sensitivity experiment-3) diurnal variation**
Here, diurnal variation of monitored aerosols and gases were analysed to investigate the gas-to-particle
conversion of $NH_3$ in the model. We analyzed the simulation results of the sensitivity experiment-3. The diurnal
variations in aerosol species and gaseous precursors are controlled mainly by thermodynamic gas-to-particle
partitioning, PBL mixing, emission and deposition processes, along with vertical and horizontal advection
(Meng et al., 2018). Figure 10 presents the diurnal variation of $NH_3$, $Cl^-$, $NO_3^-$, $SO_4^{2-}$, $NH_4^+$, HCl, $SO_2$, HONO,
and $HNO_3$ concentrations in MARGA (Fig. 10a) and model (Fig. 10b). In the measurements (Fig.10a), between
19:00 and 9:00 h, a decrease in $NH_3$ was accompanied by an increase in $NH_4^+$ ($r$ -0.38), which coincided with
higher $Cl^-$ ($r$ -0.45) concentrations, while an increase in $NH_3$ coincided well with decreased $NH_4^+$ and $Cl^-$.
Correlation of $Cl^-$ with $NH_4^+$ shows a strong positive relation of $r = 0.98$, pointing to the chloride being the
dominant neutralizing agent for $NH_4^+$ at the observational site (Acharja et al., 2021). Also, day and night
formation of HONO and $HNO_3$ showed significant inverse correlations with $NH_3$ concentrations of $r$ -0.55 and
$r$ -0.45, respectively. Furthermore, HONO formation showed a strong positive correlation of $r = 0.68$ with
$NH_4^+$ concentration and $r = 0.69$ with $Cl^-$ concentration. Low night-time temperature acts as a driver for aerosol
formation, and the presence of $NH_3$ in the IGP may play a crucial role in promoting the hydrolysis of $NO_2$ and
other heterogeneous reactions, resulting in explosive HONO and aerosol formation (Ge et al., 2019). However,
in detail, the role of $NH_3$ and HONO formation in accelerating the secondary aerosol formation via



heterogeneous reaction pathway requires further study. Our observations indicate that HCl, along with HONO
and HNO$_3$, might play a significant role in determining the chemistry of NH$_3$ and other aerosols, similar to other
studies Acharja et al. (2021) and Fu et al. (2019).
However, after including HCl emissions from trash burning (Sharma et al., 2019) in the model (Fig.
10b), the diurnal variation in NH$_3$ is found to be controlled mainly by strong linear relation of $r = 0.81$ with Cl$^-$,
$r = 0.79$ with SO$_2$ and NH$_4^+$, $r = 0.48$ with NO$_3^-$ and inverse relation $r = -0.96$ with HCl concentration. It should
be noted that other than trash burning, various industrial plants around Delhi contribute to HCl emissions but are
not considered in this emission inventory (Jaiprakash et al., 2017). In the model, a high concentration of Cl$^-$,
SO$_2$, and NO$_3^-$ mainly controls the diurnal behavior of NH$_3$. Significant missing daytime HONO sources include
heterogeneous ground conversion, heterogeneous aerosol formation, soil emissions, and photochemical
production, which are entirely missing in the model studies (Lu et al., 2018). A few possibilities might
contribute to the modeled discrepancies. First, there are uncertainties in the emission inventory of the bottom-up
approach of NH$_3$, SO$_2$, and NO$_X$, since NH$_3$, SO$_2$ and NO$_X$ concentrations are overestimated in the model. Also,
accurate industrial sources of HCl emission need to be included in the future. Work aimed at improving NH$_3$
emissions by considering agricultural statistics on fertilizer use and animal number distribution is currently
under development as part of the Global Challenges Research Fund (GCRF), South Asian Nitrogen Hub
(SANH). Second, the low gas-to-aerosol partitioning of SO$_2$ to sulfate in the WRF-Chem model, along with the
overestimation of nitrate, can result in the discrepancies of sulfate, SO$_2$, and nitrate concentration. Third, missing
HONO formation and chemical reactions catalyzed by TMI or oxidation by NO$_2$ in the MOSAIC mechanism is
usually responsible for ~ 49 % of the SO$_2$ oxidation in a fog/haze situation, and the rest via O$_3$ oxidation (51 %)
pathway is low in the model. Correct gas-to-aerosol partitioning of HNO$_3$ in the model significantly diminishes
daytime nitrate and sulfate formation, which also obstructs the heterogeneous reaction of SO$_2$ with NO$_2$ thus
leading to a positive NH$_3$ bias during wintertime in Delhi.

**4. Conclusions**

Ground-based measurements of NH$_3$, trace gases, and water-soluble ions in PM$_{2.5}$ were made during the WIFEX
campaign from 19 December 2017 to 21 January 2018 at Indira Gandhi International Airport (IGIA) in the Indo-
Gangetic Plain (IGP), where a large amount of ammonia is emitted because of agricultural and other activities.
The averaged NH$_3$ and NH$_4^+$ concentrations measured by the MARGA instrument for the entire period were
28.20 and 36.96 µg m$^{-3}$ compared with 56.75 and 14.71 µg m$^{-3}$ as estimated by the WRF-Chem model,
respectively. The bivariate polar plot of the gas-to-particle conversion of ammonia (NH$_4^+$/NH$_3$) shows a
sporadic high value towards the west, indicating the dominance of ammonium and chloride formation. The share
of major components of gases and particulate matter (PM$_{2.5}$) based on µeq m$^{-3}$ shows, on average, the anions in
PM$_{2.5}$ are almost exactly neutralized by NH$_4^+$ (49.54 %), with Cl$^-$ (29.69 %) as the major anion. Furthermore, the
observations show NH$_4^+$/NH$_3$ being significantly correlated ($r = 0.79$) with aerosol chloride concentration, with
a much lower correlation for other anions, suggesting that ammonium chloride formation is predominant at the
observational site. Modeled values overestimated NH$_3$ (56.75 µg m$^{-3}$) and underestimated NH$_4^+$ (14.71 µg m$^{-3}$)
concentration in the WRF-Chem model. To evaluate the performance efficiency of the model, we conducted
three sensitivity experiments in the model, sensitivity experiment-1 (default set-up without HCl emissions),



sensitivity experiment-2 with base case HCl emissions (3 × original HCl emissions of Sharma et al. (2019)), and
the sensitivity experiment-3 (3 × the base case HCl emissions) from trash burning to study its effect on the gas-
to-aerosol partitioning of $NH_3$. The sensitivity experiment-3 results in significant improvements in the
simulation of $NH_4^+$ concentration (range up to 70 μg m$^{-3}$), Cl$^-$ concentration (range up to 110 μg m$^{-3}$ ), and some
improvement in $NH_3$ observed at IGIA, Delhi.

548        Furthermore, for the first time, this study reports observed and modeled HCl concentrations that were

promoting gas-to-particle conversion of ammonia by enhancing chloride concentration, suggesting that
significant emissions of HCl into an ammonia-rich atmosphere facilitate the gas-to-particle conversion of
ammonia through $NH_4Cl$ formation. Although improvement is seen and there is a decrease in $NH_3$ gas-phase
concentration by 10 % in the sensitivity experiment-3, the model still shows a high bias for $NH_3$ concentration.
This suggests that the emissions inventory overestimates emissions of $NH_3$, along with $SO_2$ and $NO_x$, and
highlights an incorrect gas-partitioning of $NH_3$ in the model.

555        This analysis is the first in South Asia to use simultaneous measurements from MARGA of $NH_3$,

$HNO_3$, HCl, and aerosols combined with a high-resolution model to understand the coupling between gas-
particles governing ammonia concentration. Our observations indicate that excess $NH_3$ concentrations along
with daytime and night-time $HNO_3$ and HONO variability may impact secondary aerosol formation. However,
to understand atmospheric photochemistry and heterogeneous chemical processes, the study of HONO and
$HNO_3$ production mechanisms are required in the future. The accuracy of the chemical transport model is also
affected by underestimation of $SO_4^{2-}$, HONO, $HNO_3$, and HCl, along with overestimation of $SO_2$ and $NO_3^-$ and
their interaction which impacts the model performance for $NH_3$. Hence, in the future, it is necessary to evaluate
the impact of the addition of correct industrial sources of HCl emission along with appropriate emissions of
HONO, $SO_2$, $NH_3$, and $NO_x$ and the addition of missing chemical reactions catalyzed by TMI, or oxidation by
$NO_2$ in the MOSAIC mechanisms which may improve the model-measurement agreement of the gas phase
ammonia.

567        The present study suggests that the bias in $NH_3$ could be reduced by using country-specific emission

inventories of $NH_3$, which are currently under development as part of the Global Challenges Research Fund
(GCRF), South Asian Nitrogen Hub (SANH). Also, there is potential to develop top-down constraints on $NH_3$
emissions by taking inference from the satellite, model, and ground-based observations. Further examination of
the role of wintertime biomass burning on $NH_4^+/NH_3$ partitioning is also merited, as this may provide a context
further promoting $NH_4^+$ formation. Therefore, further substantial addition of appropriate emissions of the trace
gases along with advanced chemistry is suggested to address the challenges of simulating atmospheric ammonia
over the IGP region. Finally, the study highlights the importance of including the treatment of HCl and its
anthropogenic emissions in modeling activities aimed at quantifying the role of $NH_3$ as a contributor to
particulate matter.
**Data availability**
The 0.1$^°$ × 0.1$^°$ emission grid maps can be downloaded from the EDGAR website on
https://edgar.jrc.ec.europa.eu/htap_v2/index.php?SECURE=_123 per year per sector. Gridded emissions in t y$^{-1}$
on a 0.1° × 0.1° for HCl emissions can be downloaded from Mendeley data: http://dx.doi.org/10.



17632/546t9249bv.1. The model data is available at Aditya, Indian Institute of Tropical Meteorology
(IITM) super-computer and can be provided upon request to the corresponding author. The observational and
meteorological data of WiFEX are available by contacting the corresponding author.
**Author contributions**
SDG designed the research; PVP performed the WRF-Chem model simulations and led the analysis; PA and RK
contributed to data collection and its quality control and assurance; GG, RK, and PG helped with the model set-
up; PVP and SDG wrote the paper with contributions from all co-authors.
**Competing interests**
The authors declare that they have no conflict of interest.
**Acknowledgments**
We thank the Director, IITM, for his continuous support and encouragement. IITM is funded by the Ministry of
Earth Sciences (MoES), Government of India. We wish to thank the MoES for supporting the WiFEX
campaign. The lead author's fellowship was supported by the National Supercomputing Mission (NSM)
program grant at C-DAC, and Ph.D. fees are covered by the Natural Environment Research Council (NERC) of
UK Research and Innovation (UKRI)-Global Challenges Research Fund (GCRF), South Asian Nitrogen Hub
(SANH), and we are grateful to the Executive Director and the Director-General of C-DAC and the SANH
Director and Chair of the Executive Board. We acknowledge the availability of CPCB-$NO_x$, $NO_2$, and $O_3$ data
from the CPCB web portal (https://app.cpcbccr.com/ccr, last access: 1 December 2021). We wish to
acknowledge the National Center for Atmospheric Research is sponsored by the National Science Foundation.

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





**FIGURE CAPTIONS**

**Figure 1. (a) Observed average diurnal variation in $NH_3$ and $NH_4^+$ with its (b) meteorological parameters during the sampling period (right).**

**Figure 2. (a) Wind rose diagram from the observational site, bivariate plots of (b) $NH_3$ concentration (c) Ammonium concentration (d) HCl concentration (e) Chloride concentration and (f) Partitioning ratio of $NH_4^+/NH_x$ in relation to wind speed (m s$^{-1}$) and direction.**

**Figure 3. (a) Comparison of observed and simulated diurnal variation in $NH_3$ and $NH_4^+$ with its (b) meteorological parameters during the sampling period.**

**Figure 4. (a) Temporal variation in the daily mean $NH_3$ concentration, (b) $NH_4^+$ concentration, and (c) $NH_x$ concentration for MARGA measurements (black line) and model (Redline).**

**Figure 5. Share of major components of gases and particulate matter ($PM_{2.5}$) based on the mean concentrations during WiFEX (share according to µeq m$^{-3}$).**

**Figure 6. Neutralizing effect between $Cl^-$, $NO_3^-$ and $SO_4^{2-}$ as the anions and aerosol neutralization ratio (ANR).**

**Figure 7. Relationship between observed $NH_4^+/NH_x$ with (a) $NH_4^+$ and (b) Chloride concentration and Relationship between modeled $NH_4^+/NH_x$ with (c) $NH_4^+$ and (d) Sulfate concentration at the IGIA site during the study period (red line denotes regression line).**

**Figure 8. Box-Whiskers plot for secondary inorganic aerosols, trace gases, and meteorological parameters from the observations and simulated by the model for the three different sensitivity experiments at Delhi.**

**Figure 9. Diurnal variation in the mean (a) $NH_3$ concentration (b) $NH_4^+$ concentration and (c) $NH_x$ concentration for MARGA measurements (black), sensitivity experiment-1 (red dotted), sensitivity experiment-2 (red dash) and sensitivity experiment-3 (red solid).**

**Figure 10. Average diurnal cycles of ammonia with $Cl^-$, $NO_3^-$, $SO_4^{2-}$, $NH_4^+$, HCl, $SO_2$, HONO, and $HNO_3$ of (a) measured (MARGA) and (b) modeled (sensitivity experiment-3).**





**TABLES**

**Table 1. Performance of statistics of Correlation coefficient ($r$) of $NH_4^+/NH_x$ with all other gases and aerosols**

| Gases and Aerosols | MARGA Correlation coefficient ($r$) with $NH_4^+/NH_x$ ratio | Model Correlation coefficient ($r$) with $NH_4^+/NH_x$ ratio |
|---|---|---|
| Ammonia ($NH_3$) | -0.57 | -0.58 |
| Sulfur dioxide ($SO_2$) | 0.46 | -0.30 |
| Hydrogen chloride (HCl) | 0.14 | - |
| Nitrous acid (HONO) | -0.24 | -0.23 |
| Nitric acid ($HNO_3$) | 0.33 | 0.46 |
| Ammonium ($NH_4^+$) | 0.70 | 0.67 |
| Chloride ($Cl^-$) | 0.79 | - |
| Sulfate ($SO_4^-$) | 0.09 | 0.77 |
| Nitrate ($NO_3^-$) | 0.13 | 0.57 |

**Table 2. Model performance statistics for $NH_3$, $NH_4^+$ and $NH_x$ concentration at IGIA, Delhi from WRF-Chem simulations and the MARGA**

| Species | Sensitivity experiment-1 | | Sensitivity experiment-2 | | Sensitivity experiment-3 | |
|---|---|---|---|---|---|---|
| | Correlation coefficient ($r$) | Normalised Mean Bias (NMB) | Correlation coefficient ($r$) | Normalised Mean Bias (NMB) | Correlation coefficient ($r$) | Normalised Mean Bias (NMB) |
| $NH_3$ | -0.58 | 1.38 | -0.60 | 1.29 | -0.65 | 1.13 |
| $NH_4^+$ | 0.45 | -0.61 | 0.75 | -0.40 | 0.76 | -0.03 |
| $NH_x$ | 0.69 | 0.12 | 0.70 | 0.22 | 0.70 | 0.39 |






**Table 3. Summary of ammonia, ammonium, and total ammonia concentration (daily mean ± standard deviation) in MARGA, sensitivity experiment-1, sensitivity experiment-2, and sensitivity experiment-3**


| Species | MARGA | Sensitivity experiment-1 | Sensitivity experiment-2 | Sensitivity experiment-3 |
|---|---|---|---|---|
| $NH_3$ concentration (μg m$^{-3}$) | 20 ± 8.52 | 50.19 ± 11.79 | 48.18 ± 11.31 | 44.94 ± 10.80 |
| $NH_4^+$ concentration (μg m$^{-3}$) | 35.94 ± 17.73 | 13.89 ± 3.04 | 21.44 ± 6.65 | 34.58 ± 15.21 |
| $NH_x$ concentration (μg m$^{-3}$) | 56.69 ± 17.15 | 164.08 ± 13.29 | 69.63 ± 16.66 | 79.52 ± 23.70 |
































**Figure 1**

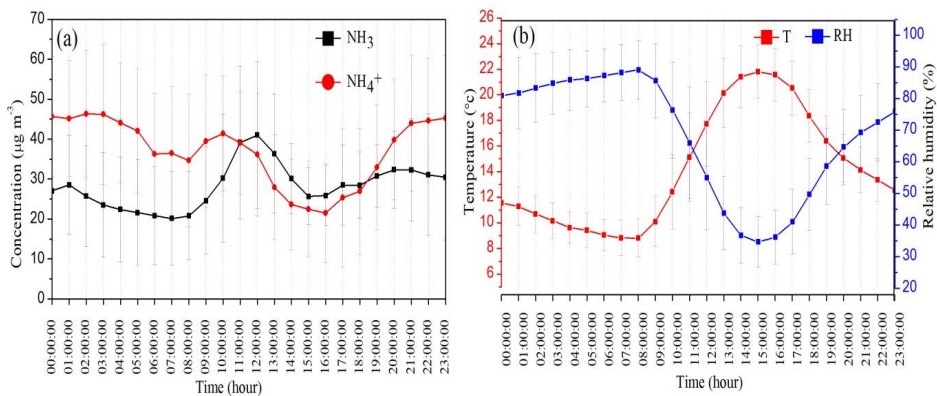


**Figure 2**

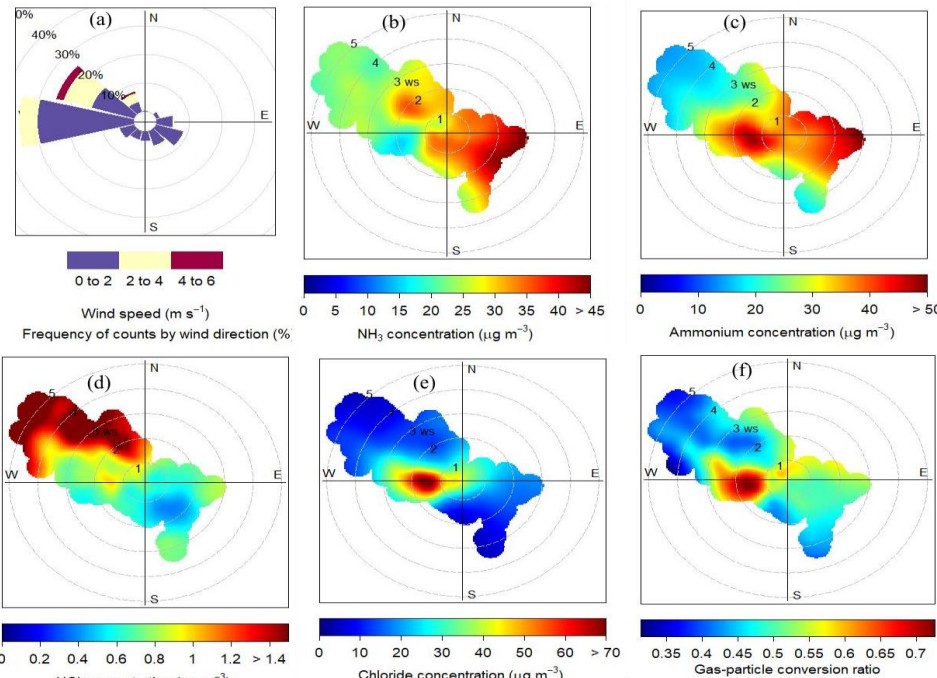











**Figure 3**

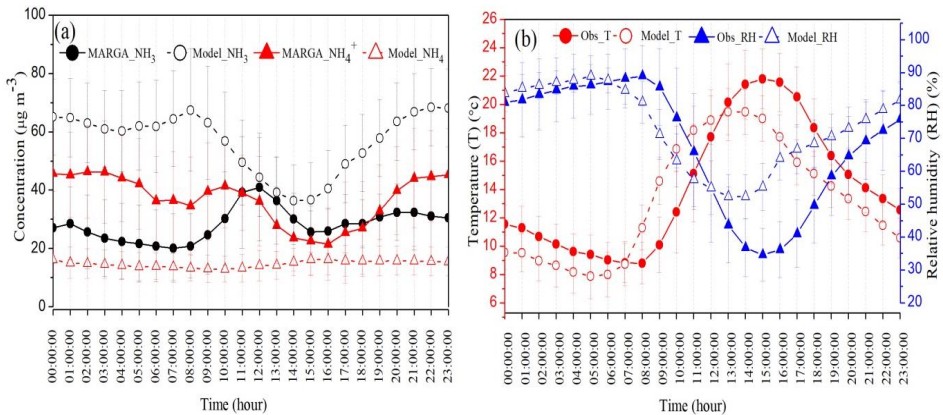


**Figure 4**

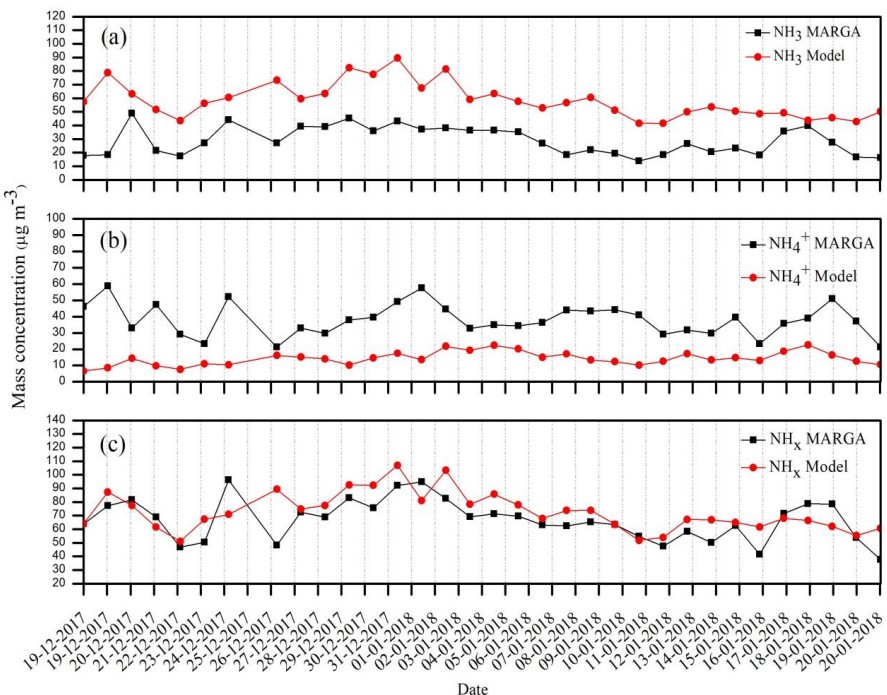








**Figure 5**

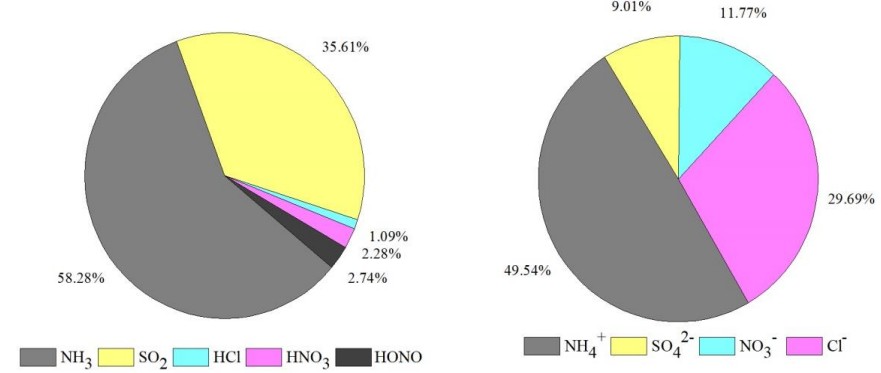

**Figure 6**

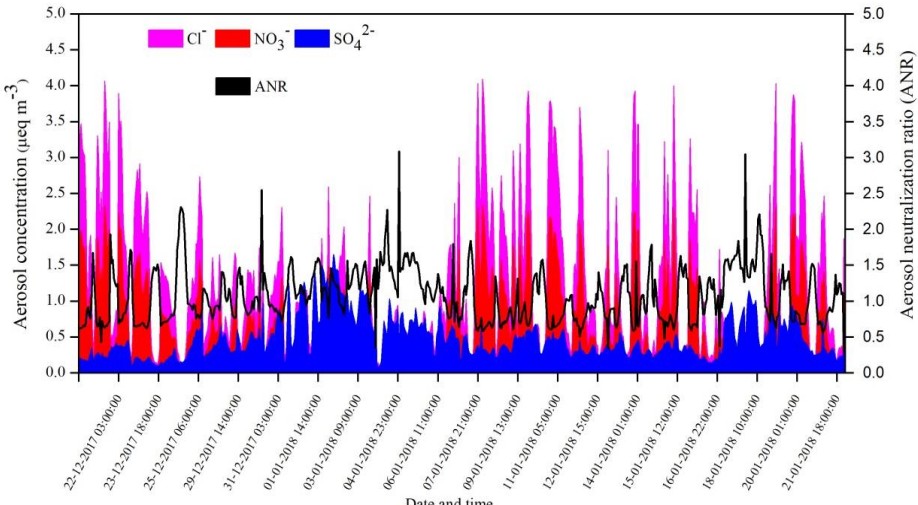





**Figure 7**

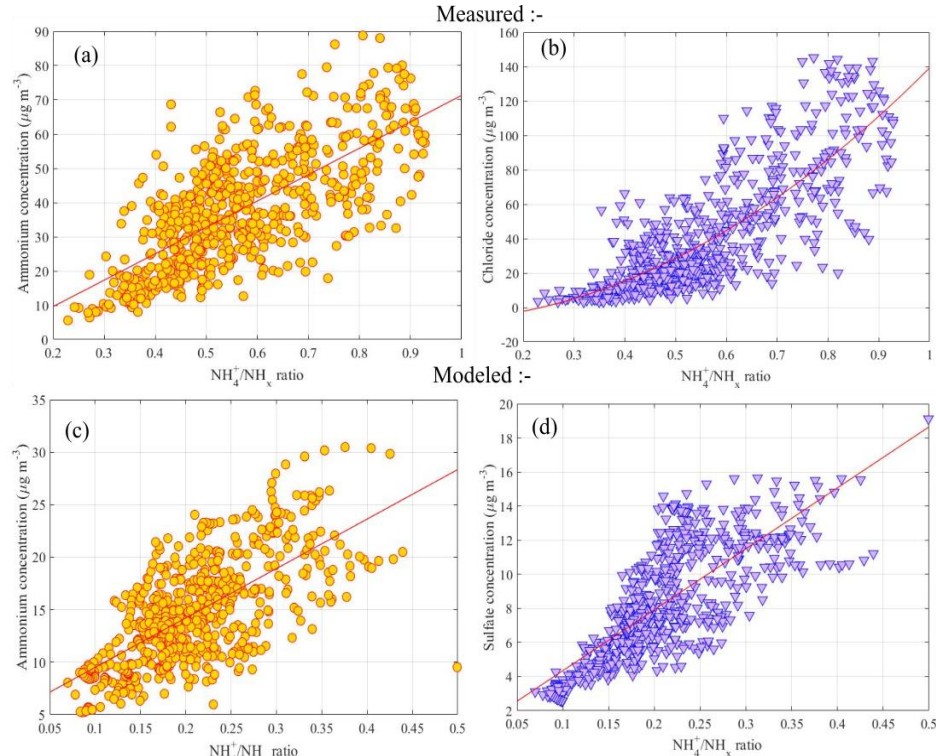





**Figure 8**

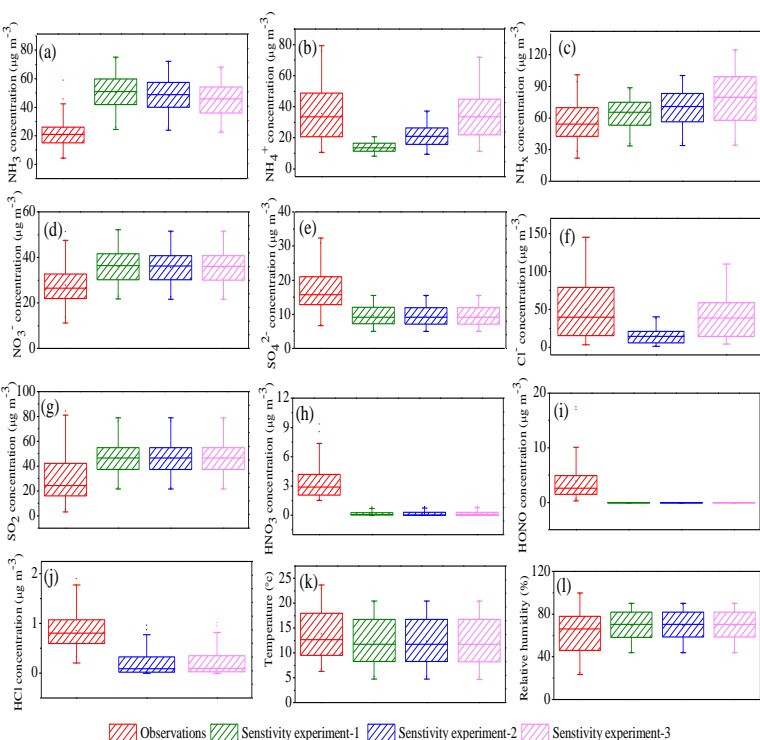




**Figure 9**

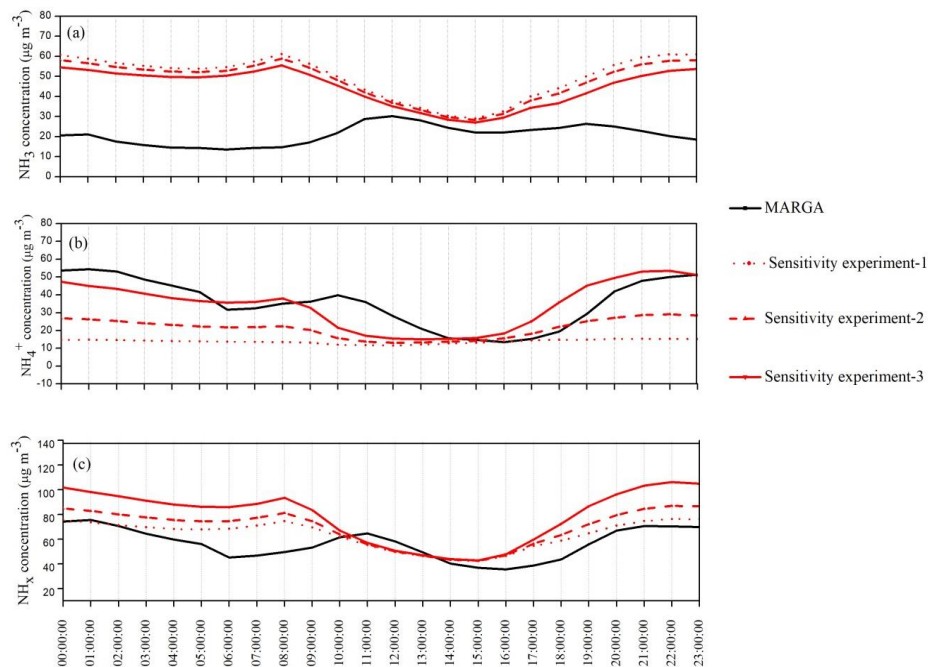























**Figure 10**

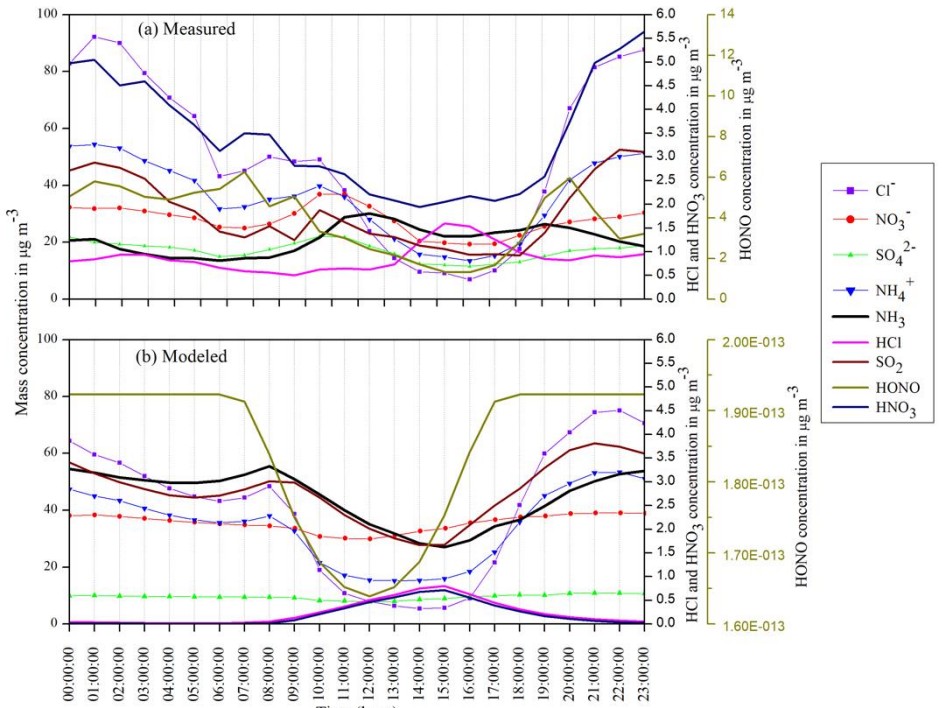
