# Peer review of "Chloride (HCl/Cl-) dominates inorganic aerosol formation"

_Atmospheric Chemistry and Physics, 2022_

## Author Comment (AC1)

Responses to the Manuscript ACP-2022-237: Hydrochloric acid emission dominates inorganic aerosol formation from ammonia in the Indo-Gangetic Plain during winter.

Dear Editor-in-Chief:

We hereby submit the revised version of our manuscript (ACP-2022-237).

We greatly appreciate the Editor-in-Chief and two referees for providing highly insightful and constructive comments, which have substantially improved the clarity of our manuscript. Our main focus of this study is to understand the capabilities of the model in simulating $NH_3$, $NH_4^+$, and $NH_x$, and to understand the possible causes behind the discrepancies between the simulations and observations. The influence of $HCl/Cl^-$ chemistry in the modeling gas-to-particle partitioning of ammonia is compared with the observations and the results are reported. Some of the aspects of measurements need to be understood more deeply in future work, such as dew evaporation, $SO_2$ oxidation pathways, and the synoptic scale events when $Cl^-$ and $NO_3^-$ were low. Chen et al. (2022), Gunthe et al. (2021), and Acharja et al. (2021) have already worked on thermodynamic modeling and chloride chemistry, its composition, and growth. However, we focus on the modeling results and comparison with observations to make our study more impactful. We have carefully addressed all the comments. Please see below our point-to-point responses in blue (our comments) and red text (major changes/additions) and refer to the revised manuscript.

In our attached documents, we have made the following major revisions:

Result and discussion sections are rearranged, focusing on the modeling and measurement comparison to understand the mismatches. The new sub-sections are framed as follows:-
- 3.1 Comparison of temporal variation in $NH_3$, $NH_4^+$, and total $NH_x$ using WRF-Chem and MARGA
- 3.2 Gas-to-particle partitioning
- 3.3 Influence of $HCl/Cl^-$ chemistry in WRF-Chem
- 3.4 Comparison of the temporal variation in $NH_3$, $NH_4^+$, and $NH_x$ using WRF-Chem ($HCl/Cl^-$) and MARGA
- New Fig.4 is added in the revised manuscript to understand the driver of measured $NH_4^+$ and the role of aqueous chemistry, we plotted the fraction of $HCl/Cl^-$ as a function of $NH_4^+$ concentration and RH.
- To maintain the focus of the manuscript, a few sections are omitted such as HONO chemistry and its discussion.
- Further details on the MARGA measurement methods and QA/QC are added 2.1.1 and 2.1.2.
- One sensitivity case study was performed in the model by reducing $NH_3$ emissions to fix the $NH_x$ overestimation and Fig. 10 is included.
- Streamlined the manuscript focussing on the role of $HCl/Cl^-$ chemistry in giving rise to the discrepancies between the observations and the model simulations.

We hope you find our manuscript suitable for publication and look forward to hearing from you.

**Replies to the comments from the anonymous Referee #1**

The authors are thankful to the reviewer for a thorough review and for raising several interesting and valid points that provided us an opportunity to clarify several aspects of this manuscript and improve it overall. Below are our responses to the reviewer.

*Comment1:* comments and revisions to the manuscript when applicable. This manuscript expands on the analysis of previously published work by using WRF-Chem to attempt to model the observations described in Acharja et al 2020 and Acharja et al 2021. The authors need to clarify that the same dataset was the topic of the previous papers that thoroughly describes the aerosol trends and chemistry. These papers are cited in a few places but adding a sentence that explicitly makes the connection needs to be included. To keep the differentiation between the current and previous papers, and to make the modelling results more impactful, the analysis needs be more clearly framed using the model to understand the mismatch in the model and the observations of $NH_3/NH_4^+$ partitioning even though NHx was well reproduced. The main result highlighted in the title of the manuscript is important but the modeling is not needed to determine that $NH_4Cl$ is an important form of particulate ammonium. The manuscript would greatly benefit from a more focused and streamlined discussion.

*Response:* We agree with the reviewer. The major revisions in the result and discussions are reframed and discussed focusing on the modeling aspect. We agree that analysis of the ion balance from the MARGA measurements is sufficient to show the importance of $NH_4^+$ and $Cl^-$ in $PM_{2.5}$. However, our analysis goes further to show the need to extend widely used modelling approaches to recognize the major significance of chloride sources in South Asian context.

Please refer to section 2.1 and section 3 in the revised version of manuscript.

And the title modified is

"Chloride ($HCl/Cl^-$) dominates inorganic aerosol formation from ammonia in the Indo-Gangetic Plain during winter: Modeling and comparison with observations."

*Comment2:* The discussion of the data used could use some clarification. While there is in-depth discussion of how $PM_1$ and $PM_{2.5}$ were separated there is no discussion of how the gases were separated and measured. This detail is important to add even if it appears in the other paper referenced describing the methods. Were both $PM_1$ and $PM_{2.5}$ used in the modeling? Which size cut are you using for the comparison of the model and measurement?

*Response:* Section 2.1 is modified accordingly to describe the methods (Section 2.1.1) used. In addition concerning the rational and data used, we have added:

"We have used only $PM_{2.5}$ inorganic water-soluble components and the gaseous measurements (available from both the $PM_1$ and $PM_{2.5}$ MARGA collection systems). Since $NH_4^+$ with the three major anions: $Cl^-$, $NO_3^-$ and $SO_4^{2-}$ constituted 97.3 % of the total measured ions in $PM_{2.5}$

(Acharja et al., 2020), we consider these four significant ions in our present study. In contrast, the remaining ionic species (i.e., $Na^+$, $K^+$, $Mg^{2+,}$ and $Ca^{2+}$) contributed only about 3 % of the total measured ions and were neglected as it would not impact our present study significantly (Acharja et al., 2020)."

"For consistency with the $PM_{2.5}$ MARGA measurements, we have chosen 3-bins according to simulated aerosols size (0.04–0.156 µm; 0.156–0.625 µm; 0.625–2.5 µm) in accordance with the WRF-Chem aerosol size distribution."

*Comment3:* On Line 123 you state that the $PM_{2.5}$ impactor was cleaned regularly – what about the $PM_1$ impactor?

*Response:* Yes, both $PM_1$ and $PM_{2.5}$ impactors were cleaned regularly. For clarification, we explicitly included the phrase :

"The $PM_1$ and $PM_{2.5}$ impactors were typically cleaned fortnightly to remove any material that may stick on the surface and inlets of the impactors"

*Comment4:* The discussion of detection limits (Line 132). For the ozone data, you state that data below the LOD were rejected – did you just omit them and treat as NA or replace with zero or fill with the DL? Were any of the MARGA data below the LOD, how did you treat those data?

*Response:* Yes, for the observation data including MARGA, we omitted the values below LOD. We have omitted them and treated them as NA. Thus, to enhance clarity, we have further addressed the potential for bias in the remaining dataset by noting that only a small fraction of the datasets were excluded by being below or above the instrument detection limit and maxima as follows:

"The lower detection limits (LODs) of the species monitored by MARGA were mentioned in Acharja et al. (2021). It shows that concentrations of species like $Cl^-$, $NO_3^-$, $SO_4^{2-}$, $NH_4^+$, $SO_2$, and $NH_3$ were always higher than LODs during the winter period. But, concentrations of species like $Na^+$, $K^+$, $Ca^{2+}$, $Mg^{2+}$, HCl, HONO, and $HNO_3$ were sometimes below LODs, but the fraction of it was less than ~10 % of the total observation period. We have omitted these values and treated them as NA. As the fraction of observational hours is less and these species contribute much less to the $PM_1$ and $PM_{2.5}$ mass concentrations, we believe below LODs values would not significantly deviate our results. The quality of the data obtained was then checked using the ion-balance method. As an additional quality check, the ratio of the sum of cations to anions (neq $m^{-3}$) was used as an indicator for the viable data. We have checked the cation-to-anion ratio of each hourly sample expressed in the unit of neq $m^{-3}$. We accepted only those values near to unity and rejected those not within the 10 % error bar limit. Based on this evaluation method, overall, for the campaign, the ratio was near unity (1.06 for $PM_1$ and 0.96 for $PM_{2.5}$). Excellent charge balance between anions and cations measured by the system also

confirms that there are no significant contamination issues associated with the aerosol measurements. Values in slight excess of unity may indicate the presence of formate and acetate in the aerosol, which MARGA does not measure. Further detail on the quality control of MARGA can be found in Acharja et al. (2020)"

"For data quality of CPCB, we omitted all those observed values which fell below LOD of the instrument (2 µg m$^{-3}$ for NO$_x$ and 4 µg m$^{-3}$ for O$_3$) (Technical specifications for CAAQM station, 2019) and above 500 µg m$^{-3}$ for NO$_x$ and 140 µg m$^{-3}$ for O$_3$ and treated them as NA at a given site. For the NO$_x$ and O$_3$ datasets, only a small fraction of data (2 %) were outside the instrument operating ranges specified"

*Comment5:* How was the MARGA calibrated?  Were blanks taken?  I didn't see this information in the other papers referenced from your group. The internal calibration standard just accounts for dilution not verification of the concentrations measured.  Did you do comparisons with other sampling techniques to help validate the MARGA data?

Each analysis of the MARGA system was calibrated using the Lithium bromide (LiBr) internal standard containing 320 µg L$^{-1}$ lithium (Li$^+$ ) and 3680 µg L$^{-1}$ bromide  (Br$^-$). Unlike in the URG AIM instrument, for example, in the MARGA the internal calibration standard does not just act as a dilution check. It provides an actual ongoing calibration. The concentrations of the different compounds are derived via the specific conductivities of the various ions compared to those included in the internal standard. This is explained in detail by Oms et al. (1996) and Thomas et al.(2009). The quantification relies on 100 % efficient gas and aerosol collection, which is well established for the system.  The main periodic instrument calibrations relate to establishing the instrument background and checking for potential contamination (more important for measuring low concentrations than those reported here) and periodic calibration of the mass flow controllers.  Excellent charge balance between anions and cations measured by the system also confirms that there are no significant contamination issues, at least associated with the aerosol measurements.

Because the MARGA dataset has already been published (Acharja et al., 2020; 2021), however, we do not feel that this manuscript needs to reiterate the full measurement approach.

**"2.1.2 Quality assurance/quality control (QA/QC) of MARGA**

To ensure the observation's accuracy and check the data's quality, we have taken all the precautionary measures during the study. The eluents, absorption, and regenerant solutions were prepared with minimum manual intervention. The operational parameters like anion, cation conductivity, SJAC heater temperature, column oven temperature, and airflow were regularly monitored to keep them within the safe limit. In addition to these, before injection of each sample into the anion and cation IC columns, the Lithium Bromide (LiBr) internal standard solution containing 320 µg l$^{-1}$ lithium (Li$^+$ ) and 3680 µg l$^{-1}$ bromide (Br$^-$) was mixed

with each sample to provide calibration of each analysis. This ensures that each analysis is calibrated and the concentration of gaseous and ionic samples are measured accurately. The $PM_1$ and $PM_{2.5}$ impactors were typically cleaned fortnightly to remove any material that may stick on the surface and inlets of the impactors. The lower detection limits (LODs) of the species monitored by MARGA were mentioned in Acharja et al. (2021). It shows that concentrations of species like $Cl^-$, $NO_3^-$, $SO_4^{2-}$, $NH_4^+$, $SO_2$, and $NH_3$ were always higher than LODs during the winter period. But, concentrations of species like $Na^+$, $K^+$, $Ca^{2+}$, $Mg^{2+}$, HCl, HONO, and $HNO_3$ were sometimes below LODs, but the fraction of it was less than ~10 % of the total observation period. We have omitted these values and treated them as NA. As the fraction of observational hours is less and these species contribute much less to the $PM_1$ and $PM_{2.5}$ mass concentrations, we believe below LODs values would not significantly deviate our results. The quality of the data obtained was then checked using the ion-balance method. As an additional quality check, the ratio of the sum of cations to anions (neq $m^{-3}$) was used as an indicator for the viable data. We have checked the cation-to-anion ratio of each hourly sample expressed in the unit of neq $m^{-3}$. We accepted only those values near to unity and rejected those not within the 10 % error bar limit. Based on this evaluation method, overall, for the campaign, the ratio was near unity (1.06 for $PM_1$ and 0.96 for $PM_{2.5}$). Excellent charge balance between anions and cations measured by the system also confirms that there are no significant contamination issues associated with the aerosol measurements. Values in slight excess of unity may indicate the presence of formate and acetate in the aerosol, which MARGA does not measure. Further detail on the quality control of MARGA can be found in Acharja et al. (2020)."

*Comment6:* In the WRF-Chem description section the description of the various methods/algorithms is hard to follow. I believe you are using MOSAIC with MTEM-MESA-ASTEM as described in Zaveri et al 2008. The authors need to revise the wording in Line 157 and Line 161 – ASTEM was new in 2008 and is not now (unless something else has been added). Instead of describing and naming the various components (MTEM, MESA, ASTEM) since they are in the cited paper, I suggest focusing on what MOSAIC does and doesn't not do well. This information is sort of in there but gets lost. Adding more recent papers that use the same WRF-Chem setup would improve the section.

*Response:* We have modified the revised manuscript taking on board these helpful comments.

We added the following explanation in the WRF-Chem description

"MOSAIC incorporates the thermodynamic and gas-particle partitioning module described by Zaveri et al. (2008). To reduce the computational cost, we selected a 4-bin MOSAIC mechanism that simulates thermodynamic equilibrium and other aerosol processes such as condensation, coagulation, and nucleation. The same mechanism has been widely used with WRF-Chem for simulations outside India (Bucaram and Bowman, 2021; Sha et al., 2019; Yang et al., 2018), but only a limited number of studies have applied it to the Indian domain to include more detailed chemistry and species (Gupta and Mohan, 2015; Jena et al., 2020; Kumar et al., 2018). The SOA formation in MOSAIC is simulated using the volatility basis set approach (Knote et al., 2015). For consistency with the $PM_{2.5}$ MARGA measurements, we have chosen 3-bins according to simulated aerosols size (0.04–0.156 μm; 0.156–0.625 μm; 0.625–2.5 μm) in accordance with the WRF-Chem aerosol size distribution."

*Comment7:* Since the focus of this manuscript are the modeling results and discrepancies between the observations and model I suggest a reorganization of the Results and Discussion section. The section should start with 3.2.1 and 3.2.2. Some description of the temporal trends can be discussed which then nicely flows to potential processes that are missing - like dew and fog (section 3.1.1) or potential issues with the current model setup (partitioning/sources/Cl chemistry, etc).

*Response:* The revised manuscript is now streamlined accordingly in the results and discussion sections as follows:-

- 3.1 Comparison of temporal variation in $NH_3$, $NH_4^+$, and $NH_x$ using WRF-Chem and MARGA
- 3.2 Gas-to-particle partitioning
- 3.3 Influence of $HCl/Cl^-$ chemistry in WRF-Chem
- 3.4 Comparison of the temporal variation in $NH_3$, $NH_4^+$, and $NH_x$ using WRF-Chem ($HCl/Cl^-$) and MARGA

*Comment8:* Line 216-230: The discussion of fog/dew should be shortened since you don't fully investigate this process and one of the other papers from this study looked at the role of fog. This discussion seems like a distraction from the main points of the paper – no need to describe the figures in the supplement just say you looked at this and details can be found there. I think these are more helpful to your main points as they processes that aren't included in the model. The discussion of dew as a potential night-time source should include Wentworth et al 2016. Wentworth, G.R., Murphy, J.G., Benedict, K.B., Bangs, E.J., Jr, J.L.C., 2016. The role of dew as a night-time reservoir and morning source for atmospheric ammonia. Atmospheric Chemistry and Physics 16, 7435–7449. https://doi.org/10.5194/acp-16-7435-2016

*Response:* We agree with the reviewer. According to the suggestions, we moved the discussion of fog/dew information to supplementary material, explaining the discrepancies of the model and the observations, while keeping it short.

"We also looked into the average diurnal profile of $NO_x$ and the $NH_3$ during dense fog events, and the details can be found in the supplement (Fig. S1 and S2 in the Supplement). It is evident that the observed daytime peak of $NH_3$ did not coincide with $NO_x$ peaks, suggesting that traffic emissions do not contribute significantly to the observed $NH_3$ rise. The observed correlation between fog water and enhanced $NH_3$ pulses is consistent with what would also be expected from the evaporation of dew (Sutton et al., 1998; Wentworth et al., 2014, 2016) (S2 in the Supplement) but is not sufficient to identify whether it is the main cause of the daytime increase of $NH_3$. In the future, measurements of the dew water $NH_4^+$ and the accumulation of dew water would be ideal for illuminating the contributing processes. The daytime increase in $NH_3$ concentration could be associated with $NH_4^+$ aerosol volatilization driven by an associated sharp change in T and RH (~ 11:00-12:00 h) (Sutton et al., 2009a, 2013) off-ground surfaces. The fastest increase in T is 12:00 h, which is indeed when $NH_3$ was at maximum concentration indicating gas-to-particle partitioning may impact the diurnal behavior of $NH_3$ at Delhi during winter (Sutton et al., 2009a, 2009b). However, in the model, because the largest increase in simulated $NH_3$ also precedes the large changes in simulated meteorological parameters, and because the simulated particulate $NH_4^+$ is flat compared to observations, simulated meteorology is ruled out as a significant contribution to high bias in simulated $NH_3$. Also, the current model does not include the bidirectional exchange of $NH_3$ with surfaces such as dew and fog water."

*Comment9:* Section 3.1.2 focuses more on understanding what the data is telling us about sources in the area, rather than how the emissions inventory might be right/wrong. Can you frame the discussion in this section differently to highlight the connection to the modeling results and getting the right $NH_3/NH_4^+$ partitioning?

*Response:* Section merged and shortended in 3.2 Gas-to-particle partitioning whereas, getting the right $NH_3/NH_4^+$ partitioning is highlighted in section 3.4 Comparison of the temporal variation in $NH_3$, $NH_4^+$, and total $NH_x$ using WRF-Chem (HCl/Cl⁻) and MARGA.

*Comment10:* Line 335: The pH dependent reaction of $SO_2$ oxidation by $O_3$ is an aqueous phase reaction. There is evidence of a heterogenous oxidation reactions on sea salt (https://agupubs.onlinelibrary.wiley.com/doi/pdf/10.1029/2006JD008207) but that is not what the authors are referring to from Seinfeld and Pandis. Please clarify the statement that begins on Line 335 and consider removing the rest of this paragraph and the following paragraph. It is sufficient to say that gas phase oxidation is much slower than aqueous phase oxidation and due to nearby sources much of the sulfur is present as $SO_2$.

*Response:* Thanks, we have corrected accordingly.

"In a normally $NH_3$-rich atmosphere, gas-phase oxidation of $SO_2$ is much slower than the aqueous phase oxidation of $O_3$, and due to nearby sources, much of the sulfur is present as $SO_2$ (Li et al., 2007) (Fig. S3 in the Supplement). This appears to be because of the slow rate of gas phase oxidation of $SO_2$. Although the atmosphere is rich in $NH_3$, in principle favoring aqueous phase oxidation via $O_3$, it appears that $O_3$ concentrations are often insufficient (mean = 36.3, median = 33.8, minimum = 26.5, and maximum = 53.9, ug $m^{-3}$ respectively) at the IGIA site (Fig. S3 in the Supplement)."

_Comment11:_ Line 342: What is limiting $SO_2$ production to sulfate? Is it just the proximity of the source to the measurement site? This statement is confusing.

_Response:_ To avoid confusion the statement is now deleted. We are unable to be quantitative in answering this question at present and refer to previous response.

_Comment12:_ Line 388-9: The statement about HCl enhancing Cl concentrations is confusing. Perhaps restate to says HCl is available for conversion to Cl.

_Response:_ We have shortened this section and the statement is now deleted to improve clarity.

_Comment13:_ Section 3.4 – This discussion would be clarified with a table defining the modeling scenarios and what the change in total HCl emissions look like: No HCl (0 tons/yr), Base Case (3x Sharma; X tons/yr), 3xBase (Y tons/yr) or something similar. These names are more intuitive than Sensitivity experiment # and make the discussion easier to follow. You could easily do this as part of table 2 or 3.

_Response:_ We agree with this comment and have incorporated the change as suggested. To interpret and discuss the sensitivity experiments, we changed the names of Sensitivity experiment in terms of its total HCl emissions originate from our defined study area to: a) No HCl (0 mol $km^{-2}$ $h^{-1}$), b) Base Case (3× Sharma et al., 2019; 24.8 mol $km^{-2}$ $h^{-1}$) and c) 3×Base (74 mol $km^{-2}$ $h^{-1}$) HCl emissions. To be responsive to the both reviewer's suggestion, we calculated the total HCl emissions (mol $km^{-2}$ $h^{-1}$) used as a input in the model and included the changed names in the Table 2 and 3 of the revised manuscript.

**Table 2. daily mean ± 1σ in gases and inorganic aerosol concentration observed (MARGA) and simulated in sensitivity test with changes in total HCl emissions (No HCl (0 mol km$^{-2}$ h$^{-1}$), Base Case HCl (24.8 mol km$^{-2}$ h$^{-1}$), and 3×Base HCl (74 mol km$^{-2}$ h$^{-1}$).**

| Species concentration (µg m$^{-3}$) | MARGA | No HCl | Base Case HCl | 3×Base HCl |
|---|---|---|---|---|
| $NH_3$ | 20 ± 8.52 | 50.2 ± 11.7 | 48.2 ± 11.31 | 44.5 ± 10.8 |
| $NH_4^+$ | 35.9 ± 17.7 | 13.9 ± 3.04 | 21.4 ± 6.65 | 34.5 ± 15.2 |
| $NH_x$ | 56.6 ± 17.1 | 64 ± 13.2 | 69.6 ± 16.6 | 79.5 ± 23.7 |
| $Cl^-$ | 50.6 ± 39.4 | - | 15.1 ± 9.65 | 40.9 ± 27.2 |
| $NO_3^-$ | 27.9 ± 8.17 | 35.9 ± 7.23 | 35.6 ± 7.05 | 35.5 ± 7.03 |
| $SO_4^{2-}$ | 17.1 ± 5.63 | 9.62 ± 2.78 | 9.56 ± 2.71 | 9.56 ± 2.71 |
| HCl | 0.86 ± 0.35 | - | 0.20 ± 0.23 | 0.22 ± 0.25 |
| $HNO_3$ | 3.43 ± 1.68 | 0.18 ± 0.21 | 0.17 ± 0.22 | 0.18 ± 0.23 |
| $SO_2$ | 30.6 ± 18.4 | 46.6 ± 12.4 | 46.7 ± 12.4 | 46.7 ± 12.4 |

**Table 3. Model performance statistics for $NH_3$, $NH_4^+$ and total $NH_x$ concentration at IGIA, Delhi from three sensitivity experiments (No HCl (0 mol km$^{-2}$ h$^{-1}$), Base Case HCl (24.8 mol km$^{-2}$ h$^{-1}$), and 3×Base HCl (74 mol km$^{-2}$ h$^{-1}$) ) and the MARGA**

| Species | No HCl | | Base Case HCl | | 3×Base HCl | |
|---|---|---|---|---|---|---|
| | Correlation coefficient ($r$) | Normalised Mean Bias (NMB) | Correlation coefficient ($r$) | Normalised Mean Bias (NMB) | Correlation coefficient ($r$) | Normalised Mean Bias (NMB) |
| $NH_3$ | -0.58 | 1.38 | -0.60 | 1.29 | -0.65 | 1.13 |
| $NH_4^+$ | 0.45 | -0.61 | 0.75 | -0.40 | 0.76 | -0.03 |
| $NH_x$ | 0.69 | 0.12 | 0.70 | 0.22 | 0.70 | 0.39 |

*Comment14:* Table 3: how do you end up with different amounts of NHx across the different sensitivity experiments? Is this a function of the change in lifetime of NH3 and NH4?

Experiment 3 with the highest HCl emissions doesn't have the lowest total NHx so that doesn't entirely make sense to me. Line 486 – total ammonia = NHx? The sentence this is apart of doesn't clearly explain why total ammonia would increase. Can you elaborate for the reader?

*Response:*

Yes, in Table 3, we have reported different amounts of $NH_3$, $NH_4^+$ and $NH_x$ with respect to the sensitivity experiments. In Table 3, the $NH_x$ value in sensitivity experiment-1 was incorrect (typo mistake). It should be 64.08 and not 164.08, which is now corrected in the revised manuscript. Please check Table 2

"Associated with these changes, total mean $NH_x$ also increased by 5.5 and 9.8 µg m$^{-3}$ in the Base Case HCl and 3×Base HCl, respectively, compared to the No HCl. This is likely due to associated increases in the atmospheric lifetime of $NH_x$ with respect to deposition as the partitioning shifted from the faster depositing gas phase to the aerosol phase. The lifetime of $NH_3$ is very short, a few hours, while that of $NH_4^+$ is 1 to 15 days (Aneja et al., 1998; Nair and Yu, 2020; Pawar et al., 2021; Wang et al., 2020).

To understand further the overestimation of total $NH_x$ by the model, we performed a sensitivity test with the HCl emissions that led to the best model/obs comparison (3×Base HCl emissions) by additionally reducing $NH_3$ emissions by a factor of 3 (-3×NH₃_EMI). Figure 10 shows the ratio of model/obs for $NH_3$ (Fig. 10a), $NH_4^+$ (Fig. 10b) and total $NH_x$ (Fig. 10c) concentration. It can be seen that the model-measurement agreement improves significantly (model/obs closer to 1) after reducing $NH_3$ emissions for all three metrics. -3×NH₃_EMI would reduce the mean $NH_3$, $NH_4^+$, and total $NH_x$ concentration by ~8.1 µg m$^{-3}$, 3.2 µg m$^{-3}$, and 11.3 µg m$^{-3}$, respectively, compared to the 3×Base HCl run. Even though reducing $NH_3$ emissions, it is still sufficient to react rapidly with the varying HCl in the sensitivity experiments contributing to an increase in $NH_4^+$. As can be seen in Fig. 10b, initially, $NH_4^+$ is somewhat lower, but it increases later and matches the 3×Base HCl run. This suggests that $NH_4^+$ formation in the model is more sensitive to changes in HCl than changes in $NH_3$ emission, while total $NH_x$ agrees well by reducing the $NH_3$ emissions. In general, CTMs have higher $NH_3$ concentration than observations, further supporting models having too much $NH_3$."

Yes, Line 486 is total ammonia = $NH_x$. To improve the clarity for the reader we have elaborated the sentence.

*Comment15:* Line 417: How confident are you in the emission inventories of $SO_2$? Is it just a chemistry issue or are there issues with the conversion of $SO_2$ to $SO_4$? And as you mention at the end of the paragraph aqueous processing is important. This seems like it might be the more likely culprit than adding these gas phase mechanisms. Have you done sensitivity studies to see how much more $SO_4$ you can get from these different oxidation pathways? Do any of the observation days have similar cloud/fog as the model? Do they agree better?

*Response:* We agree that these are good questions. However, we could not identify the main cause for $SO_2$ to $SO_4^-$ conversion without further evidence which requires additional study. Thus we restated as below in the section 3.3 to clarify concerning the present uncertainties.

"The simulated $SO_4^{2-}$ concentration (Fig. 7f) was underestimated (by ~ 7.5 µg m$^{-3}$), while gas-phase $SO_2$ Fig. 7i) was found to be overestimated by about 16 µg m$^{-3}$ in all three experiments compared with the observations. This may be caused by the fact that the drivers for typical sulfate production via OH or aqueous $H_2O_2$ oxidation pathway are likely to be wrong in the model. The missing chemistry may underly this mismatch and requires further sensitivity studies considering different $SO_2$ oxidation pathways. This requires further study, such as scenario evaluation of altered $SO_2$ emissions in the model, to examine the main pathway(s) for $SO_2$ to $SO_4^{2-}$ conversion. Measurements of OH and other radicals in Delhi are currently lacking, making it difficult to constrain the associated chemical schemes."

*Comment16:* Line 421: define TMI

*Response:* Transition Metal Ions (TMI). We have currently deleted the statement in the revised manuscript and shorten the discussion on $SO_2$ to $SO_4^-$ conversion.

*Comment17:* Line 529: $NH_3$ is a trace gas – why separate it out?

*Response:* We agree. Conclusion - section 4. is revised completely.

*Comment18:* Line 567: If $NH_x$ generally agrees then why does the emission inventory for $NH_3$ need to be adjusted? Is there a reason a sensitivity test with $NH_3$ wasn't performed to reduce emissions? For instance, your best HCl emissions (test 3) with lower $NH_3$?

*Response:* Yes, we carried sensitivity experiment by reducing $NH_3$ emissions by a factor of 3 (-3×NH$_3$_EMI) in the model for 3×Base HCl emission test.

Accordingly we have added the new Fig. 10 in the revised manuscript:

"To understand further the overestimation of total $NH_x$ by the model, we performed a sensitivity test with the HCl emissions that led to the best model/obs comparison (3×Base HCl emissions) by additionally reducing $NH_3$ emissions by a factor of 3 (-3×NH$_3$_EMI). Figure 10 shows the ratio of model/obs for $NH_3$ (Fig. 10a), $NH_4^+$ (Fig. 10b) and total $NH_x$ (Fig. 10c) concentration. It can be seen that the model-measurement agreement improves significantly (model/obs closer to 1) after reducing $NH_3$ emissions for all three metrics. -3×NH$_3$_EMI would reduce the mean $NH_3$, $NH_4^+$, and total $NH_x$ concentration by ~8.1 µg m$^{-3}$, 3.2 µg m$^{-3}$, and 11.3 µg m$^{-3}$, respectively, compared to the 3×Base HCl run. Even though reducing $NH_3$ emissions, it is still sufficient to react rapidly with the varying HCl in the sensitivity experiments contributing to an increase in $NH_4^+$. As can be seen in Fig. 10b, initially, $NH_4^+$ is somewhat

lower, but it increases later and matches the 3×Base HCl run. This suggests that $NH_4^+$ formation in the model is more sensitive to changes in HCl than changes in $NH_3$ emission, while total $NH_x$ agrees well by reducing the $NH_3$ emissions. In general, CTMs have higher $NH_3$ concentration than observations, further supporting models having too much $NH_3$.

[Figure]

**Figure 10. Comparison of ratio of model/obs in the daily mean (a) $NH_3$ concentration (b) $NH_4^+$ concentration and (c) total $NH_x$ concentration in 3×Base HCl and -3×$NH_3$_EMI scenario.**

_Comment19:_ Figure 4: This might be more effective as a ratio plot – model/obs where 1 indicates agreement. Then you could put all 3 line together.

_Response:_ We agree.

We added the following Fig.2 in revised manuscript.

[Figure]

**Figure 2. Ratio of model/obs of the daily mean NH₃, NH₄⁺ and total NHₓ concentration**

*Comment20:* Figure 6: This timeline suggests there are periods when Cl and NO₃ are low. Does the model do better predicting NH₃ and NH₄⁺ during these times? This would support your push for better HCl chemistry. The Pink and Red are too close in shade and color to clearly read the plot.

*Response:* Yes this is correct and we welcome the suggestion of the reviewer. We have added the information in section 3.2 and figure in the supplement in the revised manuscript.

"There also were certain periods where low concentrations were observed of Cl⁻ and NO₃⁻ (03-06 January 2018 and 16-17 January 2018) in Fig. 6. Comparing the model/obs for NH₃, NH₄⁺, and total NHₓ during these periods provides some degree of validation of the model where sulfur chemistry dominates the reaction with NH₃. Figure S4 (in the supplement) shows that model/obs indicates substantial variability which appears to be overestimating NH₃ (model/obs >1) while underestimating total NH₄⁺ (model/obs <1) on average in the model."

As suggested we added the updated Fig.6 and Fig. S4 in revised manuscript:-

[Figure]

**Figure 6. Neutralizing effect between Cl⁻, NO₃⁻ and SO₄²⁻ as the anions (μeq m⁻³) and aerosol neutralization ratio (ANR) where, ANR>1 indicates over neutralized (alkaline) and ANR<1 indicates under neutralized (acid) (orange bar indicates daily mean standard error)**

[Figure]

**Figure S4. Ratio of model/obs for NH₃, NH₄⁺ and NHₓ during (a) 03-06 January 2018 and (b) 16-17 January 2018**

*Comment21:* Figure 7: This figure seems unnecessary.

*Response:* Removed

*Comment22:* Figure 8: Group by component (put HNO₃ and NO3 next to each other). I also suggest leaving a space in the HCl and Cl- plots for the no HCl case to make it more obvious that there wasn't any Cl.

*Response:* Corrected the Fig. 7 as per RC1 and RC2 comments in the revised manuscript.

[Figure]

**Figure 7. Box-Whiskers plot for trace gases and secondary inorganic aerosols from the observations (MARGA) and simulated in sensitivity test with changes in HCl emissions (No HCl (0 mol km⁻² h⁻¹), Base Case HCl (24.8 mol km⁻² h⁻¹), and 3×Base HCl (74 mol km⁻² h⁻¹)) at IGIA, Delhi.**

**Anonymous Referee #2**

*Comment1:* **General Comments**

The Authors present an observational dataset of water-soluble gas-particle inorganic atmospheric species to study the impacts of HCl on aerosol compositional changes in the IGP. They find that regionally high $NH_3$ allows local sources of atmospheric HCl to substantially partition to the condensed phase through correlational analyses and implementation of HCl emissions estimates into an atmospheric chemical model that includes an equilibrium thermodynamic model. Overall, the results are an interesting case study that builds on prior observations of the unusual and substantial presence of $NH_4Cl$ in $PM_{2.5}$ in India. Generally, the components of this work are suitable and of interest to the readership of ACP. However, in its current state there are several substantial inaccuracies that need to be addressed in the interpretation of the model data and potential drivers for mismatches between models and observations. There is substantial excess material in the manuscript that does not directly drive the main thesis of the work around HCl/Cl$^-$ importance to aerosol loadings and composition that would greatly improve this work if removed. In many cases, these additional considerations are highly speculative without the necessary supporting observations. There are many technical corrections needed to improve clarity, particularly with respect to thermodynamic partitioning of the $NH_4$-$NO_3$-$SO_4$ system, but these are rather minor. Pending these major revisions, the manuscript should be suitable for publication in ACP.

*Response:* We appreciate the reviewer's generally positive assessment of the manuscript and thank you very much for reviewing our paper and for your helpful suggestions and comments. We have followed this overarching guidance and have focused the paper more strongly on the aspects related to HCl/Cl$^-$, while removing more speculative wider discussion.

Our analysis is based on the importance of missing HCl/Cl$^-$ chemistry in the model and its impact on $NH_3$, $NH_4^+$ and $NH_x$ This is necessary as a foundation for future work to understand mismatches and improve model-measurement agreement.

**Major Revisions**

Throughout the manuscript, the authors lean on the concept that ammonium salts are formed in aerosols from their gas phase precursors. While this is possible, most aerosols have sufficient liquid water that a solid salt is not the product, but rather that dynamic exchange between the gas and condensed phase sustains or alters the condensed phase composition. The writing of salt formation and neutralization reactions throughout the manuscript conflict with the use of an equilibrium thermodynamic model and some effort should be made throughout to accurately relate the relevant processes. There are many instances where this needs to be considered and they are presented in the detailed comments below, which may not comprehensively identify all occurrences. The Authors are encouraged to revise the manuscript thoroughly to address this. Given the focus of this manuscript is on an evaluation of the MARGA analyte dataset, enough information needs to be presented on its operation, calibration, and quality control to

demonstrate that high quality data was obtained. Specific QA/QC for these datasets and for instrument operation are required. Details of the MARGA from other field campaigns do not lend credibility to the QA/QC obtained here. Every move and new setup of an instrument requires a revisit and verification of functionality. Lay out your figures of merit (accuracy, precision, etc) required to operate the instrument and how these are obtained. Were background corrections done? How? And so on... See

more detailed comments for this section below.

*Response:*

The writing of salt formation and neutralization reactions were removed from the manuscript and the Section 3.2 is modified according to equilibrium thermodynamic partitioning and relevant literature are cited. In addition the fraction of HCl/Cl⁻ ratio is plotted against $NH_4^+$ and RH in revised manuscript in Fig. 4 to illustrate the links to thermodynamics

Details of MARGA methodology, calibration and QA/QC are added in the revised manuscript Section 2.1 (as also requested by Reviewer #1). Specifically we include:

[revised manuscript text omitted]

*Comment2:* Sections 3.3 and 3.5 have numerous issues and require major revision for accuracy.

*Response:* We agree and we have revised the sections 3.2 and 3.4 Section and rearranged as per RC1 suggestions. Section 3.2 is modified according to equilibrium thermodynamic partitioning by evaluating the fraction of NH$_x$ in the particulate phase (NH$_4^+$/NH$_x$) and relevant literature are cited. Specifically we include:

- Modified the theory of equilibrium thermodynamic partitioning and the reactions.
- New Fig.4 is added in the revised manuscript to illustrate the links to thermodynamics
- Shorten the discussion on local emission sources and highlighted the important details of NH$_3$ sources.
- Added the short discussion on modeling the observed synoptic scale events.
- In Section 3.4, new Fig. 8 (3.4.1) is added based on equivalent units and the discussion is modified.
- In section 3.4.2, Table 3 is modified.
- One sensitivity case study was performed by reducing the NH$_3$ emissions and the details are added.

See detailed comments below.

*Comment3:* The manuscript is long for the investigation being performed. There is substantial redundancy between some sections and in some other cases, an entire paragraph can be distilled into a single sentence. Identifying further opportunities to improve the concision of the manuscript will also improve its clarity, and raise its impact. A few places where this would be very helpful are identified, but since the role of the Reviewer is not editorial in nature, a recommendation to review the manuscript for other opportunities to reduce manuscript length is presented as a major revision.

*Response:* We have revised the sections and shorten the length foussing on key aspects of modeling.

*Comment4:* One major distraction throughout the manuscript is the speculative referencing to HONO chemistry and its role in $NH_4^+$ aerosol chemistry. This should be removed entirely from the manuscript, as there is not sufficient supporting data to enable a proper set of chemical inferences on the controls and relationships between HONO and aerosol composition presented. A couple brief comments on the linkages might be important, if there is prior data for this specific region that suggests this is the case from a detailed analysis. Otherwise, keep the manuscript focused on the influence of HCl/Cl-.

*Response:* We agree, and in accordance also with Reviewer #1 we have removed discussion of HONO chemistry in the revised manuscript and keeping the focus on the influence of HCl/Cl⁻ in modeling the fraction of gas-to-particle conversion.

Detailed Comments and Technical Corrections

*Comment5:* Page 1, Lines 34-36: This summary of the model performance is unclear and confusing. Please revise this and other aspects of the abstract after addressing all comments.

*Response:* Yes, corrected. We have modified the abstract section of the model performance in a more clear way.

"The Winter Fog Experiment (WiFEX) was an intensive field campaign conducted at Indira Gandhi International Airport (IGIA) Delhi, India, in the Indo-Gangetic Plain (IGP) during the winter of 2017-2018. Here, we report the first comparison in South Asia of high temporal resolution simulation of ammonia ($NH_3$) along with ammonium ($NH_4^+$) and total $NH_x$ (= $NH_3$ + $NH_4^+$) using the Weather Research and Forecasting model coupled with chemistry (WRF-Chem) and measurements made using the Monitor for AeRosols and Gases in Ambient Air (MARGA) at the WiFEX research site. In the present study, we incorporated Model for Simulating Aerosol Interactions and Chemistry (MOSAIC) aerosol scheme into the WRF-Chem. Despite simulated total $NH_x$ values/variability often agreed well with the observations, the model frequently simulated higher $NH_3$ and lower $NH_4^+$ concentrations than the observations. Under the winter conditions of high relative humidity (RH) in Delhi, hydrogen chloride (HCl) was found to promote the increase in the particle fraction of $NH_4^+$ (which accounted for 49.5 % of the resolved aerosol in equivalent units) with chloride (Cl⁻) (29.7 %) as the primary anion. By contrast, the absence of chloride (HCl/Cl⁻) and their chemistry in the standard WRF-Chem model results in the prediction of sulfate ($SO_4^{2-}$) as the dominant inorganic aerosol anion. To understand the mismatch associated with the fraction of $NH_x$ in the particulate phase ($NH_4^+/NH_x$), we added HCl/Cl⁻ to the model and evaluated the influence of its chemistry by conducting three sensitivity experiments using the model: No HCl, Base Case HCl (using a published waste burning inventory), and 3×Base HCl run. We found that 3×Base HCl increased the simulated average $NH_4^+$ by 13.1 µg m⁻³ and $NH_x$ by 9.8 µg m⁻³ concentration while reducing the average $NH_3$ by 3.2 µg m⁻³, which is more in accord with the measurements. Thus HCl/Cl⁻ chemistry in the model increases total $NH_x$ concentration, which was further demonstrated by reducing $NH_3$ emissions by a factor of 3 (-3×$NH_3$_EMI) in the 3×Base HCl simulation. Reducing $NH_3$ emissions in the 3×Base HCl simulation successfully addressed the

discrepancy between measured and modeled total $NH_x$. We conclude that modeling the fate of $NH_3$ in Delhi requires a correct chemistry mechanism accounting for chloride dynamics with accurate inventories of both $NH_3$ and HCl emissions."

*Comment6:* Page 2, Line 51: This sentence could use some clarification. Is the intention here to identify $NH_3$ as a $PM_{2.5}$ precursor gas? Instead of writing 'ammonium' three times, perhaps you can state that it is the typical counter ion for the three anions stated?

*Response:* We corrected the text in the section 1 in the revised manuscript.

"Ammonia is one of the important aerosol precursor gases, and ammonium ($NH_4^+$) is a major counter ion for the three anions such as chloride ($Cl^-$), nitrate ($NO_3^-$), and sulfate ($SO_4^{2-}$) contributing to $PM_{2.5}$ composition (Seinfeld et al., 2016)."

*Comment7:* Page 2, Line 70: Remove 'etc' and use 'for example' earlier in this sentence.

*Response:* This is now changed in Section 1 in the revised manuscript.

*Comment8:* Page 3, Line 111: Is this length of 1 cm correct? Seems too short to get air to an instrument to condense the aerosol inside a building. Please revise for clarity. Also, this instrument is only sampling particles? Or both gases and particles? If you have two size cuts, are these being modulated or do you have two separate channels for analysis? The sampling rate suggests that the observation intervals are 1 hr? Please clarify. This section could use a bit more organization. It is hard to tell if the measurements are close to real time or collected and analysed later?

*Response:* Indeed, this was a typo. The length of PolyTetraFluroEthylene (PTFE) inlet is 2 m which is now corrected. The instrument measures both gases and particles. It has two separate inlets (impactors) for $PM_1$ and $PM_{2.5}$. We have clarified this in the new extended description of the MARGA methodology. The observational interval is 1 hr, which we have clarified in the manuscript. The measurements are close to real time, as two sets of syringes are employed to collect the samples in which a set of syringe collects the sample and other set sends the collected samples from the previous hour for analysis and the relevant clarification is now added to section 2.1 of the revised manuscript.

*Comment9:* Page 4, Line 119: 'analyzed in the analyzer box using the ion chromatography (IC)' –Confusing and unclear. Please revise with technical description on how sample analysis was performed by IC. This is too superficially presented.

*Response:* We have replaced the statement with a more extended description in section 2.1

Detailes of the sampling method, operational principle, and calibration method are given in Section 2.1.

*Comment10:* Page 4, Line 122: 'absorbing solution' – What is this? For the gas collection? It's not

described at all. What was used? At least some basic description is needed here.

*Response:* The absorption solution is a thin film of 10 ppm $H_2O_2$ in water which continuously coats the wet rotating annualar denuder (WRD), thereby providing a sink for to allow the gases to diffuse into the aqueous film. The absorption solution strips water soluble gases from the laminar air stream, and is continuously changed, as it is pumped to the IC for chemical analysis. We have added an extended description in the revised manuscript: -

"Each MARGA sampling system consists of a steam jet aerosol collector (SJAC) and a wet rotating denuder (WRD) for collecting and measuring water-soluble inorganic particulate species and gases in the ambient air. The continuous coating of the WRD by a thin film of absorption solution (10 ppm hydrogen peroxide ($H_2O_2$)) allows the diffusion of gases into the absorption solution. By contrast, the low diffusion velocity of sub-micron particles restricts the ability of water-soluble aerosols to diffuse into the absorption solution. The absorption solution is continually changed to replace that abstracted for ion chromatography (IC) analysis of the dissolved gases. The air stream, depleted of gases by the WRD, subsequently enters the SJAC, where the steam enhances water-soluble aerosols to grow, allowing their mechanical capture in a cyclone. The aqueous solutions deriving from two cyclones (for $PM_1$ and $PM_{2.5}$, respectively) are then supplied to the IC for chemical analysis (Acharja et al., 2020). "

*Comment11:* Page 4, Line 125: While this statement to see earlier work is fine, a clear metric of what qualified as viable data by the ion-balance method and what does not still needs to be presented here.

*Response:* We have dded the information in the revised manuscript.

"The lower detection limits (LODs) of the species monitored by MARGA were mentioned in Acharja et al. (2021). It shows that concentrations of species like $Cl^-$, $NO_3^-$, $SO_4^{2-}$, $NH_4^+$, $SO_2$, and $NH_3$ were always higher than LODs during the winter period. But, concentrations of species like $Na^+$, $K^+$, $Ca^{2+}$, $Mg^{2+}$, HCl, HONO, and $HNO_3$ were sometimes below LODs, but the fraction of it was less than ~10 % of the total observation period. We have omitted these values and treated them as NA. As the fraction of observational hours is less and these species contribute much less to the $PM_1$ and $PM_{2.5}$ mass concentrations, we believe below LODs values would not significantly deviate our results. The quality of the data obtained was then checked using the ion-balance method. As an additional quality check, the ratio of the sum of cations to anions (neq m$^{-3}$) was used as an indicator for the viable data. We have checked the cation-to-anion ratio of each hourly sample expressed in the unit of neq m$^{-3}$. We accepted only those

values near to unity and rejected those not within the 10 % error bar limit. Based on this evaluation method, overall, for the campaign, the ratio was near unity (1.06 for $PM_1$ and 0.96 for $PM_{2.5}$). Excellent charge balance between anions and cations measured by the system also confirms that there are no significant contamination issues associated with the aerosol measurements. Values in slight excess of unity may indicate the presence of formate and acetate in the aerosol, which MARGA does not measure. Further detail on the quality control of MARGA can be found in Acharja et al. (2020)."

*Comment12:* Page 4, Line 128-131: $NO_x$ analyzers are standard instruments. Suggest giving the manufacturer and models of the instruments instead of these descriptions, along with time resolution and performance metrics relevant to the campaign instead of their principle of

operation.

*Response:* We have removed the sentence and added the following information in the revised manuscript.

"CPCB follows United States Environmental Protection Agency (USEPA) approved AC32M $NO_x$ and 42M $O_3$ analyser manufactured by Environment S. A. India Private Limited. We used one hour monitored $NO_x$ and $O_3$ values in our study."

*Comment13:* Page 4, Line 133: 'lowest'? There is only one detection limit for a given instrument or method. Suggest removing this to improve clarity.

*Response:* Agree and corrected.

"For data quality of CPCB, we omitted all those observed values which fell below LOD of the instrument (2 µg m$^{-3}$ for $NO_x$ and 4 µg m$^{-3}$ for $O_3$) (Technical specifications for CAAQM station, 2019) and above 500 µg m$^{-3}$ for $NO_x$ and 140 µg m$^{-3}$ for $O_3$ and treated them as NA at a given site. For the $NO_x$ and $O_3$ datasets, only a small fraction of data (2 %) were outside the instrument operating ranges specified."

*Comment14:* Page 4, Lines 138-139: This statement can likely be removed so long as it follows the criteria and rationale presented above. If it does, it is redundant with that material and makes it confusing why this is worth noting here? Or are these below the threshold criteria, but there is a clear local plume passing by the station? Please clarify why this is important to mention if the data do not meet the thresholding criteria.

*Response:* Statement removed since it is repetitive and not needed.

*Comment15:* Pages 4-5, Lines 153-166: Way too much detail on the composite modules. None of these details is used to investigate the controls on the gas-particle partitioning system.

Suggest removing altogether to reduce manuscript length or moving it to a section of SI, if the Authors wish to retain it. Given the superficial inspection of the model results, the level of detail given seems like a restatement of material presented by the model developers elsewhere.

*Response:* We have modified as per RC1-6 comments in section 2.2 in the revised manuscript. In summary we have made the following main changes:

"(Ghude et al., 2020; Kulkarni et al., 2020). This study used the Model for Ozone And Related chemical Tracers (MOZART-4) gas-phase chemical mechanism coupled with the Model for Simulating Aerosol Interactions and Chemistry (MOSAIC) aerosol scheme, that simulates $SO_4^{2-}$, $NH_4^+$, $NO_3^-$, methanesulfonate, $Na^+$, $Ca^{2+}$, $Cl^-$, carbonate, black carbon (BC), and primary organic mass (OC). Other inert minerals, trace elements, and inorganic species are lumped together as different inorganic masses. MOSAIC allows gas-to-particle formation, which includes $NH_3$, HCl, sulfuric acid ($H_2SO_4$), $HNO_3$, and methane sulfonic acid (MSA), and also includes secondary organic aerosols (SOA). Aerosol size distributions are represented by a sectional aerosol bin approach with four size bins (Georgiou et al., 2018). MOSAIC incorporates the thermodynamic and gas-particle partitioning module described by Zaveri et al. (2008). To reduce the computational cost, we selected a 4-bin MOSAIC mechanism that simulates thermodynamic equilibrium and other aerosol processes such as condensation, coagulation, and nucleation. The same mechanism has been widely used with WRF-Chem for simulations outside India (Bucaram and Bowman, 2021; Sha et al., 2019; Yang et al., 2018), but only a limited number of studies have applied it to the Indian domain to include more detailed chemistry and species (Gupta and Mohan, 2015; Jena et al., 2020; Kumar et al., 2018). The SOA formation in MOSAIC is simulated using the volatility basis set approach (Knote et al., 2015). For consistency with the PM$_{2.5}$ MARGA measurements, we have chosen 3-bins according to simulated aerosols size (0.04–0.156 μm; 0.156–0.625 μm; 0.625–2.5 μm) in accordance with the WRF-Chem aerosol size distribution."

*Comment16:* Page 5, Line 163: Sentence makes sense without 'however'. There is no contrast required here. This is simply a fact. Delete. Look for other sentences that start like this (e.g. thus, therefore, on the other hand, etc). These tend to confuse the point of sentences when used where they are not needed. If you can remove these from a sentence and not lose the purpose of your writing, the simplification should be preferred.

*Response:* We have deleted the sentence and modified as per reply to RC#2 15

*Comment17:* Page 5, Line 192: Based on other online IC measurements of atmospheric composition, this seems like too many significant digits? Please confirm by providing the figures of merit in the methods that show these values are consistent with instrument performance, or revise to provide appropriate values here.

*Response:* We have modified, now limiting to three significant figures throughout the paper. We have also shortened to provide only mean value.

"while MARGA measurements indicate an average $NH_3$ and $NH_4^+ \pm 1\sigma$ mass loading of 28.2 $\pm$ 12.4 and 36.9 $\pm$ 15.1 µg m$^{-3}$, respectively."

*Comment18:* Page 6, Lines 203-205: Can the $NH_3$ be coming from volatilization off of ground surfaces or from metabolism of microbes driving surface sources with a T-dependence? See more work from Murphy and Moravek on bidirectional exchange of $NH_3$ from surfaces as substantial sources of atmospheric $NH_3$.

*Response:* We have modified in section 3.1.1 as follows:

"The daytime increase in $NH_3$ concentration could be associated with $NH_4^+$ aerosol volatilization driven by an associated sharp change in T and RH (~ 11:00-12:00 h) (Sutton et al., 2009a, 2013) off-ground surfaces. The fastest increase in T is 12:00 h, which is indeed when $NH_3$ was at maximum concentration indicating gas-to-particle partitioning may impact the diurnal behavior of $NH_3$ at Delhi during winter (Sutton et al., 2009a, 2009b)."

More research is needed to explore volatilization from the different land types which requires mesasurments of the ratio of ammonium to hydronium in surface liquid pools which controls the potential for emission, surface temperature, surface wetness, soil measurements and fluxes of $NH_3$ (Ellis et al., 2011; Massad et al., 2010; Moravek et al., 2019; Sutton et al., 2009b, 2013; Wentworth et al., 2014). We aim to look into the bidirectional exchange of $NH_3$ from the surfaces in future.

*Comment19:* Page 6, Line 207: There is more recent work explicitly investigating dew-$NH_3$ interactions since the report by Ellis et al by Wentworth and Murphy (and references therein).

*Response:* We added the reference (Wentworth et al., 2014, 2016) in the revised manuscript in section 3.1.1.

*Comment20:* Page 6, Line 225: 'indicating strong evidence' - Well no. This is evidence that fog water can act as a reservoir. What you can say is that fog water enhancing $NH_3$ pulses in the morning is consistent with what would also be expected from the evaporation of dew. If you didn't measure the dew water $NH_4^+$ and the accumulation of dew water, then you don't have strong evidence. Also, why are these surface sources missed in the discussion of model

observation mismatches later on? It seems to have been forgotten? Or were the sections written by different contributors in the Author list?

*Response:* Sorry we missed this process in the model measurements discrepancies. As per RC#1 and RC#2 suggestion we have shortened the fog discussion and made the required changes discussed above. These surface sources were missed in the discussion part, and we have now incorporated it in the revised version of the manuscript in section 3.1.1.

"We also looked into the average diurnal profile of $NO_x$ and the $NH_3$ during dense fog events, and the details can be found in the supplement (Fig. S1 and S2 in the Supplement). It is evident that the observed daytime peak of $NH_3$ did not coincide with $NO_x$ peaks, suggesting that traffic emissions do not contribute significantly to the observed $NH_3$ rise. The observed correlation between fog water and enhanced $NH_3$ pulses is consistent with what would also be expected from the evaporation of dew (Sutton et al., 1998; Wentworth et al., 2014, 2016) (S2 in the Supplement) but is not sufficient to identify whether it is the main cause of the daytime increase of $NH_3$. In the future, measurements of the dew water $NH_4^+$ and the accumulation of dew water would be ideal for illuminating the contributing processes. The daytime increase in $NH_3$ concentration could be associated with $NH_4^+$ aerosol volatilization driven by an associated sharp change in T and RH (~ 11:00-12:00 h) (Sutton et al., 2009a, 2013) off-ground surfaces. The fastest increase in T is 12:00 h, which is indeed when $NH_3$ was at maximum concentration indicating gas-to-particle partitioning may impact the diurnal behavior of $NH_3$ at Delhi during winter (Sutton et al., 2009a, 2009b). However, in the model, because the largest increase in simulated $NH_3$ also precedes the large changes in simulated meteorological parameters, and because the simulated particulate $NH_4^+$ is flat compared to observations, simulated meteorology is ruled out as a significant contribution to high bias in simulated $NH_3$. Also, the current model does not include the bidirectional exchange of $NH_3$ with surfaces such as dew and fog water."

*Comment21:* Page 6, Line 229: Guttation presented out of context like this seems extremely speculative, given the composition of the $NH_3$-surface interaction literature. What about soil or stomatal interactions? The processes of guttation is not widespread amongst plant species and no literature is convincingly cited here to suggest this is necessary to point towards as a major consideration in this work. Suggest removing and perhaps visiting some more of the major contributors to $NH_3$ bidirectional exchange present in the literature to bolster likely processes governing the observed $NH_3$ and its diurnal patterns.

*Response:* We have removed mention of guttation, which is not critical to the argument. Conversely, we have added further discussion and literature suggesting major $NH_3$ contribution from dew evaporation within the $NH_3$ bidirectional exchange in section 3.1.1.

*Comment22:* Page 7, Line 236: The arguments aren't very convincing about the morning pulse of $NH_3$. Is NHx conserved and/or increasing in the morning when integrated throughout the boundary layer? If yes, then I would agree with the strength of these conclusions more. I suspect

this isn't possible to speak directly on, so would caution the Authors to be a bit more careful in their writing here.

*Response:* We have restated the statement in the revised manuscript in section 3.1.1

Evaporation processes of dew or fog water can act as a significant night-time $NH_x$ reservoir which definitely merits further investigation and could be a potential cause for the morning rise of $NH_3$ within its bi-directional framework which is currently absent in the model (Hrdina et al., 2019; Wentworth et al., 2014, 2016). Parameterizations for $NH_3$ uptake in the model during dew/fog water formation and its evaporation within its bi-directional framework should also be a focus of future work.

*Comment23:* Page 7, Lines 247-249: This sentence is mixing a lot of things together. Why is it important to mention the proximity to Delhi here? Revise for clarity.

*Response:* We have amended and deleted the rest of this section in the revised manuscript in section 3.1.1.

*Comment24:* Page 7, Lines 252-254: How is this relevant to dairies being a source of $NH_3$? Even the disposal in waste water is less likely to be a major source of $NH_3$ compared to lagoons or active application of manure to fields, followed by volatilization. Revise for accuracy and clarity.

*Response:* By dairies we referred to the housing of dairy cows (which is linked to the dairy operations), rather the processing of milk itself. High ammonia emissions are associated with dairy cows, which are typically well-fed, with high excretion and density of animals. We have amended the statement to refer to "dairy farms", while also mentioning the range of other sources in the revised manuscript in section 3.2.

"Figure 5a shows that the highest $NH_3$ concentration was associated with the winds coming from the east and southeast of the site, where it could have been emitted from dairy farms, including animal houses, yards, and manure storage, as well as by the application to the farmland of urea and other ammoniacal fertilizers, ammoniacal wastes and ruminant urine located at this region (Hindustan Times, 2021; Leytem et al., 2018; Sherlock et al., 1994). Such sources of $NH_3$ volatilization (Hristov et al., 2011; Laubach et al., 2013) can also explain the higher concentrations of total $NH_4^+$ (and, by definition $NH_x$) for air coming from the southeast of the measurement site (Fig. 5b and d)."

*Comment25:* Page 7, Lines 254-256: This argument needs to be revisited. The bivariate polar plot takes such mixing considerations into account. Revise and draw conclusions from these plots using them as they are intended to communicate information about sources.

*Response:* We have shortened this discussion as per RC1 suggestions and modified and merged in the section 3.2 in the revised manuscript and deleted this line.

*Comment26:* In the final sentence, they are higher due to the lack of turbulent mixing, meaning that plumes from point sources are diluted to a lesser extent. Diffusion rates are a fundamental property of a gas molecule and these change with temperature or pressure, but not with wind speed.

*Response:* We have shortened this discussion as per RC1 suggestions and modified and merged in the Section 3.2 in the revised manuscript and deleted this line.

"This enhancement in the southeast region is not only affected by emissions but also by meteorology and chemistry. Thus higher $NH_3$ concentration may also be due to the lack of turbulent mixing, which restricts the dilution of plumes from local point sources at lower wind speeds (Ianniello et al., 2010)."

In addition, we have referred to the effect of turbulent mixing and its diurnal variations in section 3.4.1

*Comment27:* Page 7, Lines 257-260: This belongs in a separate analysis. If you are driving phase partitioning of $NH_4^+$ from $NH_3$, then it does not have sources in industry or power plants. Be careful with the phrasing! Suggest discussion of $NH_4^+$, $Cl^-$ and $NH_x$ to follow presentation and discussion of HCl results. OR, present the $NH_4^+$ and $NH_x$ and then make a separate section about HCl/Cl to communicate that this has explanatory power for interpreting the drivers of $NH_4^+$ formation.

*Response:* We have included Fig.4 to show the role of HCl/$Cl^-$ in driving the fraction of $NH_4^+$ with a function of RH in the revised manuscript in Section 3.2., followed by bivariate plot discussion in shortened form. We have then focused on the influence of HCl/$Cl^-$ in modeling the fraction of gas-to-particle conversion of $NH_3$.

[Figure]

**Figure 4. Fraction HCl/Cl⁻ ratio as a function of NH₄⁺ concentration (µg m⁻³) and Relative humidity (RH)**

"Hence, to understand the driver of the measured $NH_4^+$ and the role of aqueous chemistry, we plotted the fraction of the ratio of HCl to Cl⁻ ($HCl/Cl^-$) as a function of $NH_4^+$ concentration and RH in Fig. 4. Fraction of particulate phase Cl⁻ increases at high RH between 70-100 % and thus increases the $NH_4^+$ concentration. The $HCl/Cl^-$ is highly anticorrelated ($r = -0.53$) with $NH_4^+$ concentration in the presence of high RH (70-100 %), further supporting the view that HCl promotes the increase in the particle fraction of $NH_4^+$ (49.5 %) with Cl⁻ (29.7 %) the primary anion."

*Comment28:* Page 7, Lines 261-262: Thermodynamic partitioning of $NH_3$ to the condensed phase. While in PM as $NH_4^+$ it can be the free ion. It doesn't have to be a salt, although there does need to be a counter-ion to maintain charge balance. Revise.

*Response:* We have modified the section 3.2 in the revised manuscript.

*Comment29:* Page 7, Lines 264-270: This needs work. Clarify that industrial sources seem probable, then speak directly on them as known HCl emitters. There is some inaccurate writing on the interactions of HCl and $NH_3$ as well. At high enough mixing ratios, these gases will homogeneously nucleate. In other cases, the excessive quantities of $NH_3$ may drive the partitioning of HCl to the condensed phase. Neither of these processes is technically a neutralization reaction and that word should be removed from this part of the discussion.

*Response:* We have removed the neutralization reaction process from the manuscript and corrected in section 3.2 in the revised manuscript.

"Two industrial sources are located in this direction: the site is impacted by a cluster in northwest Delhi of industrial processes, such as steel pickling industries, and others include

metal finishing and electroplating, which are known to be vital HCl emitters (Acharja et al., 2021; Jaiprakash et al., 2017). Near the source, abundant quantities of $NH_3$ may drive the partitioning of HCl to the condensed phase resulting in high concentrations of $NH_4^+$ and $Cl^-$ towards the west at lower wind speeds. Thus, high $NH_4^+$ and $Cl^-$ correspond to the lowest $NH_3$ concentration region (inverse relation), which can be observed in Fig. 5a, b, and c, highlighting the importance of nearby HCl industrial sources in driving the particle fraction of $NH_4^+$ and $Cl^-$ ."

*Comment30:* Page 7, Lines 270-273: I don't understand why these statements are being made? Are they potential HCl sources? Or is this general commentary on nearby industry? What do the authors mean by 'necessary' here?

*Response:* We have deleted this statement to shorten the discussion in section 3.2 in the revised manuscript.

*Comment31:* Page 8, Line 286: 'Figure 3a shows that' is repetitive. Delete. Look for other instances of such repetitiveness to improve manuscript clarity and concision.

*Response:* We have deleted and modified in the revised version.

*Comment32:* Page 8, Line 290: The writing here starts to use 'ammonia' instead of the chemical formula used up to this point. Is this writing from another Author that has not been revised for consistency? This seems to be the case, as evidenced by the use of 'vis-à-vis' which shows up in this same section (Page 9, Line 310). Please edit the manuscript for consistency in writing throughout.

*Response:* Sorry for the mistake, we have corrected and modified in the revised version.

*Comment33:* Page 9, Line 317: 'in low $NH_3$ environments' can be removed

*Response:* We have modified section 3.2 and deleted the text 'in low $NH_3$ environments'

*Comment34:* Page 9, Line 319: 'in the gas-to-particle partitioning process to produce ammonium salts…' - I agree that the $NH_3$ and acids are transferred to the condensed phase, but a lot of recent work on particle acidity clearly demonstrates that aerosols rarely have even metastable salts present in them, so this is probably better presented as the neutralization reaction from above, in the presence of water, where a pair of non-volatile $NH_4^+$ and acid anions are formed. What is depicted in these reactions is homogeneous nucleation of salts from the

gas phase collision of $NH_3$ and the acids, which isn't the primary driver of aerosol mass composition and growth.

*Response:* Agree. We have entirely revised the Section 3.2 to describe the equilibrium process as individual equilibria between the various ions in the aqueous aerosol phase and their associated gas-phase compounds.:

"The principal inorganic chemical reactions that occur in aqueous atmospheric aerosols form pairs of non-volatile $NH_4^+$ and acid anions ($SO_4^{2-}$, $NO_3^-$, and $Cl^-$) are summarized in reactions R1 to R3 (Seinfeld et al., 1998).

$$2NH_{3(g)} + H_2SO_{4(g)} \rightleftarrows NH_4^+ + SO_4^{2-} \qquad\qquad (R1)$$

$$NH_{3(g)} + HNO_{3(g)} \rightleftarrows NH_4^+ + NO_3^- \qquad\qquad (R2)$$

$$NH_{3(g)} + HCl_{(g)} \rightleftarrows NH_4^+ + Cl^- \qquad\qquad (R3)$$

$NH_4^+$ and $Cl^-$ (R3), which are favored by low T and high RH, form a reversible equilibrium with $NH_3$ and HCl (Ianniello et al., 2011; Seinfeld and Pandis, 2016), which was the case during WiFEX. It is likely that high $Cl^-$ in Delhi resulted from gas–to-particle partitioning of HCl into aerosol water in the presence of excess $NH_3$ (R3), with aqueous phase $Cl^-$ stimulating further water uptake and jointly driving aerosol mass composition and growth through co-condensation (Chen et al., 2022; Gunthe et al., 2021)."

*Comment35:* Page 9, Lines 327-328: This is not true according to thermodynamic equilibrium theory. All $SO_4^{2-}$ in the condensed phase will be fully neutralized before any $HNO_3$ or HCl can partition. Please revise for accurate representation of the state of knowledge. All this observation communicates is that the concentrations of both gas phase precursors are substantially high enough to drive phase partitioning in this very local context. Also, the quantity of $SO_2$ is not a measure of $H_2SO_4$, so the comparison being made here is misleading and the logic of this argument needs revision.

*Response:* We agree with the need to revise these lines. As suggested by RC#1, we have now condensed this section deleting the statement in the revised manuscript in Section 3.2.

"According to thermodynamic equilibrium theory, an aqueous solution maintains charge neutralization initially by balancing $NH_3$ uptake with the uptake of sulfuric acid ($H_2SO_4$) before $HNO_3$ and HCl can partition into the aqueous aerosol; hence all $SO_4^{2-}$ in the condensed phase will be fully neutralized before any $HNO_3$, or HCl can partition (Behera et al., 2013). Typical Delhi winter conditions of excess $NH_3$, high RH, and low T favor gas-to-particle partitioning of $NH_3$"

*Comment36:* Page 9, Lines 331-333: The presented data does not demonstrate anything about rate. Revise. All that can be said is that there is a lot of $SO_2$ that has not yet been converted to sulfate.

*Response:* We have corrected the statement as per RC#1-10 and drivers for $SO_2$ to $SO_4^{2-}$ . This question requires future study, hence we have condensed the section.

"In a normally $NH_3$-rich atmosphere, gas-phase oxidation of $SO_2$ is much slower than the aqueous phase oxidation of $O_3$, and due to nearby sources, much of the sulfur is present as $SO_2$ (Li et al., 2007) (Fig. S3 in the Supplement). This appears to be because of the slow rate of gas phase oxidation of $SO_2$. Although the atmosphere is rich in $NH_3$, in principle favoring aqueous phase oxidation via $O_3$, it appears that $O_3$ concentrations are often insufficient (mean = 36.3, median = 33.8, minimum = 26.5, and maximum = 53.9, ug m$^{-3}$ respectively) at the IGIA site (Fig. S3 in the Supplement). Hence for many periods during the WIFEX campaign, $SO_4^2$ and $NO_3^-$ are very low, with the result that the $NH_4^+/NH_x$ ratio does not change appreciably when $SO_4^{2-}$ is neutralized (Table 1)."

*Comment37:* Page 9, Lines 335-341: I don't understand the logic of all this text? I'm not sure it is relevant given the limited nature of this dataset and drawing from studies in other locations? Suggest removing to reduce the length of this manuscript, since the focus is on HCl impacts on PM. These alternative mechanisms driving sulfate formation are quite speculative and not very well justified, so a major simplification should be made at the very least. It is also concerning that such speculation is allowed to occupy so much of the writing about sources of sulfate, but the contributions of the know chemistry are not visited except in passing. Suggest reversing these priorities. What can current chemistry explain? Then clearly state the link between these other studies that may make them relevant to filling in the remainder of the observations to motivate future campaigns.

*Response:* We have removed the speculative statement from the revised manuscript (Section 3.2).

*Comment38:* Page 9, Line 344: 'also low daytime' – Should this be 'lower during the daytime'?

*Response:* This point is now removed due to other shortening of Section 3.2

*Comment39:* Page 9, Lines 345-346: Only in the local observations, it will be oxidized downwind. Rephrase.

*Response:* This point is now also removed due to other shortening of Section 3.2

*Comment40:* Page 10, Lines 352-353: This is only true if the accuracy of the measurements is very good. Please propagate the error of the measurements and put error bars on this trace. Typically, the resulting error from IC measurements applied to ANR results in a cumulative error near 50 %, given the challenges in quantifying $NH_4^+$ by IC.

*Response:* Thanks for drawing our attention to this point. There was a mistake in plotting ANR for $NO_3^-$. We have now included a corrected Fig. 6 in the revised manuscript, with daily mean standard error bar plotted on the ANR in Section 3.2. As you can see now the precision of the ANR is much better than 50 %. There is no particular challenge in quantifying $NH_4^+$ on the IC used in the MARGA. This is also evident in the excellent average charge balance depicted in Fig. 3.

"The mean $\pm$ 1σ ANR value for PM$_{2.5}$ during the observed period was 0.96 $\pm$ 0.14. It ranges from a minimum of 0.35 $\pm$ 0.04 to a maximum of 2.31 $\pm$ 0.08. Higher values than unity may indicate the presence of organic acids in the aerosol, which MARGA does not measure (Acharja et al., 2020). Also, high standard error in Fig.6 indicates the possibility of uncertainties associated with the breakthrough of $NH_3$ spikes on the denuder at high concentration (~ 1 %) (Stieger et al., 2019). However, the good charge balance indicates this wasn't a major issue."

[Figure]

**Figure 6. Neutralizing effect between Cl⁻, NO₃⁻ and SO₄²⁻ as the anions (µeq m⁻³) and aerosol neutralization ratio (ANR) where, ANR>1 indicates over neutralized (alkaline) and ANR<1 indicates under neutralized (acid) (orange bar indicates daily mean standard error)**

*Comment41:* Page 10, Line 353: 'utterly' - remove. The timeseries shows that this is not a universal truth. Figure 5 demonstrates that, on average, this was the case, so suggest redirecting

this commentary towards that figure. The ANR actually demonstrates substantial variability, which appears to be muted here due to the very large number of datapoints put into the statistical evaluation. It would be worthwhile to comment on the range observed as well. There also seem to be synoptic scale events where all of the nitrate and chloride are evaporated from the condensed phase? Seems to be a missed opportunity in the case studies that followed to learn something insightful about drivers of their partitioning.

*Response:* We have corrected and refer to our extended answers to RC2-40 and RC1-20, which now draw attention to the periods with low $NO_3^-$ and $Cl^-$ on 4-6 and 17-18 January.

*Comment42:* Page 10, Line 357: 'conversion rate' - This is not a rate. It is a ratio. This simply evaluates the equilibrium distribution of the $NH_x$ pool between the two phases. The model assumes equilibrium, where the rate of formation is equal to the rate of loss in a dynamic system.

*Response:* We agree and have modified Section 3.2 starting with model-measurement comparison as follows:

"We investigated the ability of the model to accurately describe the gas-to-particle partitioning of the measurements (MARGA) by evaluating the fraction of total $NH_x$ in the particulate phase ($NH_4^+/NH_x$) (Ellis et al., 2011; Wang et al., 2015) for which statistical values are summarized in Table 1. The correlation coefficient (r) indicates an inverse relationship of $NH_4^+/NH_x$ with $NH_3$ for both MARGA and model (r = -0.57, -0.58, respectively). A strong correlation of the MARGA ratio $NH_4^+/NH_x$ with the dominant anion concentration ($Cl^-$: r = 0.79) was observed. However, the measurement shows a poor relationship between $SO_4^{2-}$ and $NH_4^+/NH_x$ followed by $NO_3^-$, which is probably due to very low concentrations that do not change $NH_4^+/NH_x$ significantly even when $SO_4^{2-}$ and $NO_3^-$ are neutralized (see Fig. 6). By contrast, the model shows a strong correlation between $NH_4^+/NH_x$ with $SO_4^{2-}$ concentration (r = 0.77). MARGA indicates high particulate fractions of $NH_4^+$ and $Cl^-$, while the modeled composition is dominated by $NH_4^+$ and $SO_4^{2-}$. This mismatch is due to the complete absence of $Cl^-$ chemistry in the standard model. The measured $NH_4^+/NH_x$ suggests that anthropogenic HCl may be promoting this increase in particle fraction of $NH_4^+$ and $Cl^-$ via partitioning into the aerosol, deprotonating in the aerosol water, followed by $NH_3$ partitioning and being protonated by the ionization of the strong electrolyte HCl (Chen et al., 2022; Gunthe et al., 2021)."

*Comment43:* Page 10, Line 358: 'Previous studies have reported…' - Reported is not sufficient here. State exactly what the intention you have in using this ratio here is. You are evaluating the model against the observations, but it isn't clearly stated why the way you are doing it is the right way to go about things. What is the point of doing this for sulfate in the model, but chloride in the measurements? I cannot follow the logic. Probably this whole sentence can be deleted and the references can be moved to the end of the prior sentence. Stating what previous

studies have reported, when that is already a term in the prior sentence, doesn't really add value here, but it does add confusion when trying to follow this section.

*Response:* We have shortened and amended the text. Please refer to our reply to RC2-42 and the previous comment.

*Comment44:* Page 10, Line 367: Again. This is not a rate, which evaluates a change in concentration over time due to a given chemical process. Here you are observing the change in the particle fraction. That's it. I agree the HCl is promoting this increase in the particle fraction of $NH_4^+$. Please clarify.

*Response:* We have corrected as requested. Please refer to our reply to RC2-40-42.

*Comment45:* Page 10, Line 370: 'the reactions of ammonia with HCl' - Maybe, but I don't see any evaluation of the gas product (Kp) of $NH_3$ and HCl being performed here to suggest these are undergoing homogeneous condensation as $NH_4Cl$ to the $PM_{2.5}$. Isn't it more likely that the HCl is partitioning into the aerosol, deprotonating in the aerosol water, followed by the ammonia partitioning and being protonated by the ionisation of the strong electrolyte HCl? This is the largely agreed upon process for aerosol growth and partitioning that has been communicated in the aerosol pH community over the past 10 years.

*Response:* We agree. Please refer to our replies to RC2-34 to 35. We have deleted the homogeneous condensation reaction statement to avoid confusion and modified the statement in the revised manuscript in the section 3.2. The focus of this work is on model-measurement agreement and we are not showing existence of a homogeneous condensation reaction. Hence we have not evaluated the Kp.

*Comment45:* Page 10, Line 372-374: That is not what the anticorrelation between $NH_4^+$ and HONO means. It's could be generating nitrite in the aerosol due to the excess $NH_3$ available. The correlation here is also very poor, so this statement is highly speculative and should be removed, along with the references. A general comment on the inverse relationship, a clear statement of any speculation, then the need for further study is the most that seems appropriate here, given the limited nature of the dataset. The recommendation of the Reviewer is to remove all of this discussion as it is not relevant to the manuscript and there is not enough supporting measurements to really justify further comment.

*Response:* We have deleted the entire discussion of HONO in the revised manuscript.

*Comment46:* Page 10, Line 381: The plots of $NH_4$ versus $NH_4/NH_x$ in both the measurements and model does not make sense to present. This relationship has to be the case, since the terms

are internally dependent, and the statements in the manuscript, plus entries in the table communicate everything shown in these figure panels anyways. This is an opportunity to streamline the manuscript. Suggest removing a) and C) from Figure 7 unless substantive discussion is added to explain the value that is not already clear from the existing statements.

*Response:* We agree and have deleted Fig. 7, noting that RC1 suggested the same.

*Comment47:* Page 10, Line 385: It doesn't react. It's an aqueous solution that obtains charge neutralization through mole balance of $NH_3$ uptake before $HNO_3$ can partition. Revise all of this for equilibrium thermodynamic partitioning accuracy.

*Response:* This is amended in Section 3.2 of revised manuscript.

*Comment48:* Page 11, Line 393: Section 3.4 is way too long and needs to be substantially condensed and simplified. Suggestions follow on some ways to do this, but likely that further gains can be made.

*Response:* We hav modifed and condensed it in new Section 3.3. in particular:

- Removing the discussion on HONO and $SO_4^{2-}$ oxidation mechanism.
- Shortening the section focussing on the model scenarios after adding $HCl/Cl^-$ in the model. Modified table no. 2 and 3 with details on model scenario with respect to change in HCl emissions in model.

*Comment49:* Page 11, Line 400: '7th to 16th' does not follow journal guidelines for presentation of dates.

*Response:* We have corrected in revised manuscript (section 3.3):

"We further conducted three scenario simulations for the period 7-16 January 2018 (10 days)"

*Comment50:* Page 11, Lines 402-404: Why state the revised work twice? Instead, state who did the revision and why that is important to build upon in your work. Perhaps try to simplify the statements here and combine this sentence with the one that follows.

*Response:* We agree and have revised Section 3.3 accordingly:

"We tested the three sensitivity experiments named: No HCl (0 mol km$^{-2}$ h$^{-1}$), Base Case HCl (3× Sharma et al., 2019; 24.8 mol km$^{-2}$ h$^{-1}$), and 3×Base HCl (74 mol km$^{-2}$ h$^{-1}$) scenario, reflecting adjustments which are consistent with the more recent upward adjustments in the amount of waste burned in landfills by Chaudhary et al. (2021) and also to reflect additional industrial HCl sources not accounted for in the inventory."

*Comment51:* Page 11, Line 406: Is the base case 'experiment-2'? The organization of the model runs here is getting confusing and could use some work to improve clarity. Rename these or something.

*Response:* We have renamed and reorganized as also requested by Reviewer #1 and #2, as follows:

"We tested the three sensitivity experiments named: No HCl (0 mol km$^{-2}$ h$^{-1}$), Base Case HCl (3× Sharma et al., 2019; 24.8 mol km$^{-2}$ h$^{-1}$), and 3×Base HCl (74 mol km$^{-2}$ h$^{-1}$) scenario, reflecting adjustments which are consistent with the more recent upward adjustments in the amount of waste burned in landfills by Chaudhary et al. (2021) and also to reflect additional industrial HCl sources not accounted for in the inventory."

*Comment52:* Page 11, Lines 409-416: Example of simplification, where this can be a single sentence: 'Increasing the emissions of HCl in the model partition more NH$_3$ to the condensed phase, due to its high concentrations, reaching mass loadings of NH$_4^+$ and Cl$^-$ of 70 and 110 ug/m3, respectively'

*Response:* We thank the reviewer for this suggestion. We have modified section 3.3:

"As can be observed from Fig. 7(a-c), increasing the HCl emissions (Fig. 7g) in the model partitions more NH$_3$ to the condensed phase due to its high concentrations, reaching maximum mass loadings of NH$_4^+$ and Cl$^-$ of 70 and 110 μg m$^{-3}$, respectively, in the 3×Base HCl scenario, while increasing the total mean NH$_x$ concentration by 15 μg m$^{-3}$ compared to the No HCl run presumably reflecting the longer residence time of NH$_4^+$ for near-surface air measurements."

*Comment53:* Page 11, Lines 419-425: This is pure speculation. Remove. State instead that missing chemistry may underly the mismatch and move on. These findings from these studies may not be relevant to the Delhi observation site.

*Response:* We agree. This is amended in section 3.3:

"This may be caused by the fact that the drivers for typical sulfate production via OH or aqueous H$_2$O$_2$ oxidation pathway are likely to be wrong in the model. The missing chemistry may underly this mismatch and requires further sensitivity studies considering different SO$_2$ oxidation pathways. This requires further study, such as scenario evaluation of altered SO$_2$ emissions in the model, to examine the main pathway(s) for SO$_2$ to SO$_4^{2-}$conversion. Measurements of OH and other radicals in Delhi are currently lacking, making it difficult to constrain the associated chemical schemes."

*Comment53:* Page 12, Lines 426-432: Why all this speculation? The drivers of typical sulfate production are likely wrong in the model with so much pollution, meaning that OH or aqueous $H_2O_2$ is not being simulated correctly either. If the fundamentals can't be verified, there doesn't seem to be much justification for exploring or commenting on these other mechanisms. At a minimum, remove the comment about nucleation, as this is a very minor contributor to sulfate mass loading increases.

*Response:* We have shortened the section and refer to our reply to RC2-52.

*Comment54:* Page 12, Lines 435-436: Given the focus of this manuscript on HCl partitioning, suggest removing all discussion of HONO. No model can get this right in such polluted regimes and it is not diagnostic in an investigation of HCl partitioning unless you are drilling down into the thermodynamics.

*Response:* As noted above, we have removed discussion of HONO. Sections are rearranged focussing on the influence of HCl/Cl$^-$ in model.

*Comment55:* Page 12, Lines 438-439: Not true. The effective aerosol pH will determine the gas fraction of $HNO_3$ observed. You also have not evaluated the total of the two species and compared them to the model, as you do for NHx. It could mismatch simply because the HCl/Cl system is important and not included.

*Response:* To address this point, we have modified section 3.4 of the revised manuscript.

"The gas fraction of observed $HNO_3$ will be determined by aerosol pH and liquid water content based on $NH_3$ and $NO_3^-$ availability (Nenes et al., 2020). The over-prediction of $NH_3$ concentration in the model compared with the observations generates more $NO_3^-$ (and simultaneously reduces $HNO_3$), with the total fraction of $HNO_3 + NO_3^-$ (THNO$_3$) concentration in the model also exceeding the observed THNO$_3$, which is more strongly affected by reducing the $NH_3$ emissions in the model (Fig. S5 in the Supplement). On average, THNO$_3$ reduced by only 0.38 µg m$^{-3}$ in 3×Base HCl compared to the No HCl run. But reducing $NH_3$ emissions by a factor of 3 (-3×NH$_3$_EMI) in the 3×Base HCl scenario reduced mean THNO$_3$ by a further 4.71 µg m$^{-3}$. The extent of partitioning and accumulation of $NH_4NO_3$ depends on T, aerosol water, pH, as well as $NH_3$ availability (Nenes et al., 2020). Our model simulations find that the presence of HCl/Cl$^-$ does not significantly alter THNO$_3$ but that the excess $NH_3$ with missing chloride chemistry is a major contributor and will lead to mismatches in the model between measured simulated gas and particulate matter concentrations."

[Figure]

**Figure S5. Box-Whiskers plot for THNO₃ (HNO₃ + NO₃⁻) concentration from observations (red) and simulated by the No HCl (green), 3×Base HCl (pink) and -3×NH₃_EMI (orange) at IGIA, Delhi.**

*Comment56:* The following sentence on over-representation of $NH_3$ is more explanatory. More $NO_x$ in the model should generate more $HNO_3$, so one would expect the total of $HNO_3^+$ nitrate in the model to exceed the observed sum.

*Response:* We have modified section 3.4 in the revised manuscript and refer to our reply to RC2-55..

*Comment57:* Page 12, Lines 444-445: The extent of the partitioning and accumulation of $NH_4NO_3$ depends on T, aerosol water and pH, as well as other constituents. If the HCl/Cl-system is a major contributor, its missing will lead to mismatches.

*Response:* We agree. The text is modified in the section 3.4 in the revised manuscript:

"The extent of partitioning and accumulation of $NH_4NO_3$ depends on T, aerosol water, pH, as well as $NH_3$ availability (Nenes et al., 2020). Our model simulations find that the presence of HCl/Cl⁻ does not significantly alter THNO₃ but that the excess $NH_3$ with missing chloride chemistry is a major contributor and will lead to mismatches in the model between measured simulated gas and particulate matter concentrations."

*Comment58:* Page 12, Line 446: HONO rarely can partition to $PM_{2.5}$ aerosol that is dominated by traditional inorganics (the pH is far too low). Remove from the manuscript.

*Response:* We agree, and this is removed from the revised manuscript.

.

*Comment59:* Page 12, Lines 458-459: This is already stated in the methods. No need to repeat here.

*Response:* This is now removed from the revised manuscript.

*Comment60:* Page 13, Line 466: Why not call these:

No HCl (experiment-1)

HCl base case (experiment-2)

3xHCl (experiment3)?

Would be much easier to follow.

And why wasn't a simulation with reduced NH3 and 3xHCl in the model performed? The

NHx simulation in this run is substantially higher and the reason for that is not really

clear? Shouldn't this be conserved across all three model runs?

*Response:* Thank you very much!

We named it as : a) No HCl (0 mol km$^{-2}$ h$^{-1}$), b) Base Case HCl (3× Sharma et al., 2019; 24.8= mol km$^{-2}$ h$^{-1}$) and c) 3×Base HCl (74 mol km$^{-2}$ h$^{-1}$). To be responsive to the both reviewer's suggestion, we calculated the total HCl emissions (mol km$^{-2}$ h$^{-1}$) used as a input in the model

As noted in reply to reviewer #1 (RC1-14,18), this can be explained by following:-

"This is likely due to associated increases in the atmospheric lifetime of NH$_x$ with respect to deposition as the partitioning shifted from the faster depositing gas phase to the aerosol phase. The lifetime of NH$_3$ is very short, a few hours, while that of NH$_4^+$ is 1 to 15 days (Aneja et al., 1998; Nair and Yu, 2020; Pawar et al., 2021; Wang et al., 2020).

To understand further the overestimation of total NH$_x$ by the model, we performed a sensitivity test with the HCl emissions that led to the best model/obs comparison (3×Base HCl emissions) by additionally reducing NH$_3$ emissions by a factor of 3 (-3×NH$_3$_EMI). Figure 10 shows the ratio of model/obs for NH$_3$ (Fig. 10a), NH$_4^+$ (Fig. 10b) and total NH$_x$ (Fig. 10c) concentration. It can be seen that the model-measurement agreement improves significantly (model/obs closer to 1) after reducing NH$_3$ emissions for all three metrics. -3×NH$_3$_EMI would reduce the mean NH$_3$, NH$_4^+$, and total NH$_x$ concentration by ~8.1 µg m$^{-3}$, 3.2 µg m$^{-3}$, and 11.3 µg m$^{-3}$, respectively, compared to the 3×Base HCl run. Even though reducing NH$_3$ emissions, it is still sufficient to react rapidly with the varying HCl in the sensitivity experiments contributing to an increase in NH$_4^+$. As can be seen in Fig. 10b, initially, NH$_4^+$ is somewhat lower, but it

increases later and matches the 3×Base HCl run. This suggests that $NH_4^+$ formation in the model is more sensitive to changes in HCl than changes in $NH_3$ emission, while total $NH_x$ agrees well by reducing the $NH_3$ emissions. In general, CTMs have higher $NH_3$ concentration than observations, further supporting models having too much $NH_3$. A few factors might contribute to the model discrepancies for $NH_3$: there are uncertainties in the emission inventory of the bottom-up approach of $NH_3$, and the model does not currently include the bidirectional exchange of $NH_3$ with surfaces, such as dew and fog water. Also model does not have accurate industrial sources of HCl emission. Diurnal emission profiles are uncertainty for both $NH_3$ and HCl. Furthermore, gas-to-particle partitioning associated with $SO_2$ oxidation pathways in the model is not correct at present."

*Comment61:* Page 13, Lines 473-475: This should have been used diagnostically to conclude that there is something wrong with these simulations, or that the model includes some additional source of NHx into the modeled system that may be important. The sum of $NH_3$ + $NH_4^+$ should be conserved across all three model runs if the emissions are the same in each run. Does this chemistry scheme include bidirectional exchange of $NH_3$ from surfaces? Such processes (e.g. dew and fog) are discussed earlier on. Some substantial work needs to be done to understand the driver of this issue, as it undermines the reliability of the comparisons being made, if the assumption is that NHx should be conserved across all three sensitivity tests.

*Response:* There is no additional source of $NH_x$ in the simulations rather it is function of change in lifetime of $NH_3$ and $NH_4^+$. Please note our reply to RC1-14 and RC1-15 comments, where we note the effect of changing lifetime. The model chemistry does not include bidirectional exchange of $NH_3$ from the ground surfaces. We hence carried out another sensitivity experiment with reduced $NH_3$ emission to understand the driver of this issue in the revised manuscript. Decreasing $NH_3$ concentration for the best case (3×Base HCl) does not impact $NH_4^+$ concentration significantly but the $NH_x$ agrees well with the observations.

*Comment62:* Page 13, Lines 482-487: Why use percentages here? The relative difference between the various cases isn't very useful as a metric. Why not use the absolute change in mixing ratio or mass concentration? On Line 483 should 'improves' be 'increases'? And for Lines 486-487: This doesn't make sense. See prior comments. This should be conserved, based on how these tests have been described. This is also 'total reduced nitrogen', not 'total ammonia' as ammonia is only $NH_3$.

*Response:* We have mentioned the differences in terms of absolute change in mass concentration in the revised manuscript. Line no. 483 is now changed to 'increases'. Section

*Comment63:* Page 13, Line 488: Correlational analyses are weak tools for inferring chemical drivers. The model-measurement comparisons are much stronger in reaching robust conclusions. Suggest substantially reducing the content here that revisits the MARGA measurements and trends. This section is titled as those its purpose is to be comparing measurements to the model results? There are a lot of repetitive statements about the observations being made here that were already presented in other sections that can be removed.

*Response:* We have revised this section as 3.4.1 based on µeq m⁻³ and removed correlational analyses.

*Comment64:* Page 13, Line 497: All the correlations are done under what conditions? Isolated from 19:00 to 9:00? For the Cl- and NH4+, is this correlation performed after accounting for the NH4+ associated with sulfate and nitrate?

*Response:*

As mentioned in previous comment we decided to remove the discussion based on correlation analyses while just focussing on diurnal behaviour of chloride, sulphate, nitrate and ammonium in revised manuscript. New Fig. 8. is now included in the revised manuscript, as follows-

[Figure]

**Figure 8. (top) Average diurnal cycles of NH$_3$ and NH$_4^+$ concentration (µg m$^{-3}$) with mole equivalents of Cl$^-$, NO$_3^-$, SO$_4^{2-}$, NH$_4^+$, SO$_2$, HCl and HNO$_3$ (µeq m$^{-3}$) of (a) measured (MARGA) and (b) modeled (3×Base HCl run) along with its meteorological parameters (bottom).**

"Figure 8 (top) presents the diurnal variations of NH$_3$ and NH$_4^+$ (in µg m$^{-3}$) along with particulate NH$_4^+$, Cl$^-$, NO$_3^-$, SO$_4^2$, SO$_2$, HCl, and HNO$_3$ concentrations (in µeq m$^{-3}$) measured (Fig. 8a (top)) and modeled (Fig. 8b (top)) along with its meteorological parameters such as T and RH (Fig. 8 (bottom)). We adopted diurnal variation in emissions from Jena et al. (2021) based on boundary layer mixing. It can be seen in Fig. 8a (top and bottom) that a much bigger peak in NH$_3$ concentration is observed in the daytime than the modeled (despite turbulence differences), indeed suggesting a much stronger NH$_3$ in the middle of the day (11:00-01:00 h). As evaporation proceeds mainly in the morning (08:00-12:00) getting warmer, the peak is near midday (11:00-13:00 h), rather than in the afternoon (13:00-14:00 h) when warmest, similar to what was also observed in Sutton et al. (1998). Indeed, the decreasing NH$_4^+$ and Cl$^-$ during the late morning (10:00 h) corresponds to the increasing NH$_3$ peak, which reflects the fact that warming promotes the shift of aerosols to the gas phase. Ammonium decrease more than NH$_3$ during the day, as this also evaporates to form NH$_3$. Similarly, Cl$^-$ evaporates during the day since the HCl concentration increases. However, it can be seen that NO$_3^-$ and SO$_4^{2-}$ are slightly changed diurnally, inferring longer range transport perhaps, whereas HCl and Cl$^-$ are from more local sources. The diurnal variability in gases and aerosols in 3×Base HCl simulations in Fig. 8b (top) is primarily controlled by the planetary boundary layer mixing, meteorology/dispersion, environment (T and RH in Fig. 8b (bottom)), and transport. So presumably, maximum NH$_3$ at 08:00 h is due to limited turbulence/boundary layer, with dilution by mixing after 08:00 h. However, the model is able to represent well the diurnal variation of NH$_4^+$ and Cl$^-$ both in terms of amount and pattern, which was not the case in the No HCl run where NH$_4^+$ was observed to be flat in Section 1. During the hours of 09:00 and 11:00 h, when measured NH$_3$ rises, the model predicts a large decrease in NH$_3$, while during 19:00-23:00 h, when measured NH$_3$ decreases, the model predicts a large increase. Furthermore, the modeled HCl and HNO$_3$ are very low compared to the measurements, whereas SO$_2$ concentration matches well with the observations. It can be seen that NO$_3^-$ and SO$_4^{2-}$ are flat in the model. This highlights the need to develop accurate diurnal variability in NH$_3$ emissions over this region."

*Comment65:* Page 13, Lines 500-501: This does not add value to the analysis. HONO is well known to form at night from hydrolysis of NO$_2$, but that isn't the focus of this work. Suggest removing this analysis from this section as well. There are not enough observational constraints on the HONO chemistry to justify all this speculation.

*Response:* Thanks for your suggestion! Discussion of the HONO is removed in the revised manuscript.

*Comment66:* Page 14, Lines 509-510: Repeated information. Get rid of correlation coefficients. Focus on HCl/Cl- system interactions with NHx. Get rid of the rest.

*Response:* We agree and have amended the revised manuscript.

*Comment67:* Page 14, Lines 513-515: Remove.

*Response:* Removed.

*Comment68:* Page 14, Line 517: Repeating chemical formulas. Delete.

*Response:* Deleted.

*Comment69:* Page 14, Line 520: Partitioning and chemical transformations are two different things. This needs to be described better throughout the manuscript.

*Response:* We mentioned in revised manuscript in start of section 3.2.

"We investigated the ability of the model to accurately describe the gas-to-particle partitioning of the measurements (MARGA) by evaluating the fraction of total $NH_x$ in the particulate phase ($NH_4^+/NH_x$) (Ellis et al., 2011; Wang et al., 2015)."

*Comment70:* Page 14, Lines 522-525: The case for why this is relevant to this study is not clear and seems irrelevant. It seems highly speculative, at best, based on the current framing of the discussion and operation of the model.

*Response:* The sentence has been deleted and modified in section 3.4.

*Comment71:* Page 14, Conclusions: Rewrite to reflect on revised contents of manuscript after major revisions.

*Response*: The conclusions have been updated overall in the light of all the changes to the manscript, including more strongly focusing on the $NH_3$, $NH_4^+$, HCl, Cl$^-$ system.

*Comment72:* Page 25, Figure 2 caption: Do not capitalize ammonium and chloride here.

*Response:* Corrected.

*Comment73:* Page 25, Figure 5 caption: Was $SO_2$ treated as a divalent molecule in this analysis? That does not seem appropriate. It is not $H_2SO_4$.

*Response:* Yes, $SO_2$ was treated as a divalent molecule in this analysis. We have converted the mass concentration ($\mu g\ m^{-3}$) of $SO_2$ to $\mu eq\ m^{-3}$ considering its molecular weight and charge. We agree that it is not $H_2SO_4$, but this approach is relevant to consider the potential effects of $SO_2$ oxidation.

*Comment74:* Page 25, Figure 7 caption: Panel b does not look like a linear fit? Regression equations and values should be presented on the plots and the details of the regression (least square, linear, error-weighted, etc.) given in the caption here. This is insufficient in its current state.

*Response:* We removed this figure as suggested by RC1.

*Comment75:* Page 26, Table 1: Are correlations for HCl here actually meaningful? It's obvious that HCl

is being partitioned into the aerosol, why look at the HCl using this approach? In a fresh plume that mixes at the observation site, I would expect a strong negative correlation between HCl and NH4/NHx. This campaign wide statistical analysis for looking at local point source chemistry is confusing and, while this may have been informative for the authors to go through all these comparisons, they are not really well justified. To the Reviewer, the thing to take away here is that the chloride is completely missing in the model, but explains a lot of the variability of NH4/NHx in the observations. Suggest revising this section of the analysis to simply focus on that point and remove some of this more distracting analyses.

*Response:* Thank you for pointing out for removing unwanted parameters. As suggested, we intend to focus only on $NH_3$ and secondary inorganic aerosols such as $NH_4^+$, $Cl^-$, $SO_4^{2-}$ and $NO_3^-$. We have revised the table with major findings which focusses on missing chloride chemistry in the model. Also we have found negative correlation between fraction of $HCl/Cl^-$ and $NH_4^+$ and revised the section 3.2 accordingly.

**Table 1. Performance statistics of correlation coefficient ($r$) of $NH_4^+/NH_x$ with $NH_3$ and aerosols ($NH_4^+$, $Cl^-$, $SO_4^{2-}$, and $NO_3^-$)**

| Gases and Aerosols | MARGA | Model |
|---|---|---|
| | Correlation coefficient ($r$) with $NH_4^+/NH_x$ ratio | Correlation coefficient ($r$) with $NH_4^+/NH_x$ ratio |

| | | |
|---|---|---|
| Ammonia ($NH_3$) | -0.57 | -0.58 |
| Ammonium ($NH_4^+$) | 0.70 | 0.67 |
| Chloride ($Cl^-$) | 0.79 | - |
| Sulfate ($SO_4^-$) | 0.09 | 0.77 |
| Nitrate ($NO_3^-$) | 0.13 | 0.57 |

*Comment76:* I don't understand why the poor correlation observation between sulfate and $NH_4/NH_x$ is not being commented on? Is it because sulfate is so small that the NH4/NHx ratio doesn't change appreciably when sulfate is neutralized?

*Response:* From the charge balance in Fig. 3 and 6 and 8, it can be seen that while $SO_2$ concentration is a major component after $NH_3$ concentration, Cl is the dominant factor in determining aerosol chemistry for $NH_4^+/NH_x$. What we see in an ordinary $NH_3$-rich atmosphere gas-phase oxidation of $SO_2$ is much slower than aqueous phase oxidation via $O_3$ and due to nearby sources much of the sulfur is present as $SO_2$ (Li et al., 2007), This appears to be because of the slow rate of gas phase oxidation of $SO_2$. Although the atmosphere is rich in $NH_3$, in principle favouring aqueous phase oxidation via $O_3$, it appears that $O_3$ concentrations are often insuffient at the IGIA site (mean = 36.3, median = 33.8, minimum = 26.5 and maximum = 53.9 ug m$^{-3}$) compared to the nearest CPCB (RK_puram station) site. Hence for many periods during the WIFEX campaign, $SO_4^{2-}$ and $NO_3^-$ are very low, with the result that the $NH_4^+/NH_x$ ratio does not change appreciably when $SO_4^{2-}$ is neutralized. These aspects are included into the revised text of section 3.2.

*Comment77:* Page 28, Figure 1: The symbol for Celsius appears to be lower case? The left panel y-axis should be 'Mass concentration'. Why are time stamps displayed with minutes and seconds? Makes the figure very busy and these meaningless quantities distracting. X-axis label should be 'Time of Day' instead of 'Time' in a diurnal plot.

*Response:* Deleted the Fig. 1 due to repetition and as suggested by RC1.

*Comment78:* Page 28, Figure 2: Panel letters need to be moved. Why not state the ratio actually being calculated for the gas-particle conversion ratios instead of these words? For the frequency % in panel a, can this information be moved to the figure caption?

*Response:* As per the RC1 suggestion and RC2-Comment27, we decided to keep short discussion on sources only keeping the discussion highlighting $NH_3$, $NH_4^+$, $Cl^-$, and $NH_x$

[Figure]

**Figure 5. Bivariate plots of mean (a) NH₃ concentration (b) NH₄⁺ concentration (c) Cl⁻ concentration and (d) total NHₓ concentration in relation to wind speed (m s⁻¹) and direction.**

*Comment79:* Page 29, Figure 3: Same issue with time notation, axis labels, and temperature unit notation as Fig 1.

*Response:* Corrected.

[Figure]

**Figure 1. (a) Comparison of observed and simulated average diurnal variation in (a) meteorological parameters such as Temperature (T in °c) and Relative humidity (RH in %) and (b) NH₃ and NH₄⁺ concentration (µg m⁻³) during the sampling period (bar indicates mean standard deviation of each hour).**

*Comment80:* Page 29, Figure 4: Can the dates be presented in equal intervals and only the first dates for a given year listing that value? This information is not particularly important to the figure interpretation, so can be substantially simplified. The '-3' superscript also seems to be quite large?

*Response:* Corrected and modified as per RC1 suggestions and refer to Fig. 2 in the revised manuscript.

*Comment 81:* Page 30, Figure 6: Same issue with dates on the x-axis here. Try to simplify.

*Response:* Corrected and modified as per RC1 and RC2 suggestions. Refer to response to RC2 40 comment. Fig. 6 is modified in revised manuscript.

*Comment82:* Page 32, Figure 8: Label each plot with the species name instead of on the y-axis. Put one label for all the concentration plots on the left, after removing the HONO panel, then keep the y-axis labels for T and RH (although these can likely also be removed since this data is shown convincingly in other figures). The HONO simulations returning what appears to be zero values, suggests further that this analysis does not belong in this work. Simplify and remove. In panel (j) shouldn't a green line at zero be plotted for Experiment-1 results to keep all the panels consistent?

*Response:* Corrected and modified as per RC1 and RC2 suggestions , refer to reply to RC1 20 comment. Fig. 7 is modified in revised manuscript.

*Comment82:* Page 34, Figure 9: Remove HONO traces and axis.

*Response:* Corrected the figure 8 based on µeq m⁻³ in revised manuscript.

**References :**

[revised manuscript text omitted]

---

## Author Response (AR2)

**Replies to the comments from the anonymous Referee #1**

We greatly appreciate the Editor-in-Chief (Eleanor C. Browne) and Referee #1 for providing highly insightful and constructive comments, which have substantially improved the clarity of our manuscript. Please see below our point-to-point responses in blue (our comments) and red text (revisions) and refer to the revised manuscript.

*Comment#1:* Overall, the manuscript is significantly improved. My only major comment is that the conclusion section is hard to follow – a higher-level summary that succinctly makes the key points that are described in the main text would be more effective.

We are very thankful to the Referees for the time and effort that you have put into reviewing the previous version of the manuscript. We ensure that each comment has been addressed carefully and that the paper is revised accordingly.

*Comment#2:* I suggest:

Cutting the sentences from 478-483 (The model predicted average…to…not incorporated into the model"). Then cutting the end of the paragraph beginning at Line 490.

*Response:* Deleted in the revised manuscript.

At the beginning of the next paragraph remove "further" (line 494).

*Response:* Removed in the revised manuscript.

Then:

The revised model shows that by adding HCl emissions more NHx was partitioned to the condensed phase improving agreement with the observations. 3×Base HCl was able to represent well the diurnal variation of NH4+ and Cl- both in terms of amount and pattern with improved NMB for NH3. Additional sensitivities tests in changing NH3 emissions (reduction by a factor of 3) in the 3xBase HCl also improved the NH3, NH4, and NHx concentrations. [key last sentence beginning on 505]. These results high light the need to include correct industrial sources of HCl emissions along with appropriate emissions of NH3 to reduce biases in NHx.

Use current text:[Developing the appropriate NH3 emissions using country-specific emission inventories, which are currently under development as part of the Global Challenges Research Fund (GCRF), South Asian Nitrogen Hub (SANH). Also, there is potential to develop top-down constraints on NH3 emissions by taking inference from the satellite,

model, and ground-based observations.] Challenges remain in simulating NH3 as a contributor to particulate matter due to temporal factors in ammonia peaks including the role of fog and dew where more work is needed. This work also suggests model improvements to SO2 oxidation pathways could improve NHx partitioning.

*Response:* Conclusion section is corrected in the revised manuscript as follows:-

**Conclusion:**

"In this study, we have evaluated for the first time in South Asia the performance of a chemical transport model (WRF-Chem) in modeling $NH_3$, $NH_4^+$, and total $NH_x$, by comparing against the WiFEX measurements (MARGA). In daily means, we find $NH_3$ is significantly overestimated by the model, $NH_4^+$ was underestimated while simulated total $NH_x$ agreed well with the measurement, indicating incorrect gas-to-particle partitioning along with missing chemical process may impact this mismatch in the model. The ability of the model to accurately describe the gas-to-particle partitioning of the MARGA was evaluated by the fraction of total $NH_x$ (= $NH_3$ + $NH_4^+$) in the particulate phase ($NH_4^+/NH_x$). A strong relation of MARGA $NH_4^+/NH_x$ was observed with dominant anion ($Cl^-$) (r = 0.79), whereas the standard model showed a strong correlation between $NH_4^+/NH_x$ with dominant anion ($SO_4^{2-}$) (r = 0.77), pointing to the missing chloride ($HCl/Cl^-$) chemistry in the model.

We incorporated $HCl/Cl^-$ emissions in the model and conducted three sensitivity experiments of varying HCl emissions, named as No HCl (0 mol km$^{-2}$ h$^{-1}$), Base Case HCl (3× Sharma et al., 2019; 24.8 mol km$^{-2}$ h$^{-1}$) and 3×Base HCl (74 mol km$^{-2}$ h$^{-1}$) run. The revised model shows that by adding HCl emissions more $NH_x$ was partitioned to the condensed phase improving agreement with the observations. 3×Base HCl was able to represent well the diurnal variation of $NH_4^+$ and $Cl^-$ both in terms of amount and pattern with improved NMB for $NH_3$. Additional sensitivities tests in changing $NH_3$ emissions (reduction by a factor of 3) in the 3×Base HCl also improved $NH_3$, $NH_4^+$, and $NH_x$ concentrations. We find excess $NH_3$ along with longer lifetime of $NH_4^+$ may act as a controlling driver for $NH_x$ overestimation in the model. These results highlight the need to include correct industrial sources of HCl emissions along with appropriate emissions of $NH_3$ to reduce biases in $NH_x$. Developing the appropriate $NH_3$ emissions using country-specific emission inventories, which are currently under development as part of the Global Challenges Research Fund (GCRF), South Asian Nitrogen Hub (SANH). Also, there is potential to develop top-down constraints on $NH_3$ emissions by taking inference from the satellite, model, and ground-based observations. Challenges remain in simulating $NH_3$ as a contributor to particulate matter due to temporal factors in ammonia peaks including the role of fog and dew where more work is needed. This work also suggests model improvements to $SO_2$ oxidation pathways could improve $NH_x$ partitioning.

*Comment#3:* Minor comments:

Line 56: remove analyses

*Response:* Removed in the revised manuscript.

Line 75: remove "to its alkaline nature" – this would be an issue globally; could replace with diverse (sources)

*Response:* Removed in the revised manuscript.

Line 143: add "and" between anion and cation instead of the comma

*Response:* Added "and" in the revised manuscript.

Line 149: change to past tense; may have stuck on…

*Response:* Modified in the revised manuscript.

Line 247: remove the before NH3

*Response:* Removed in the revised manuscript.

Line 420: decreases

*Response:* Added in the revised manuscript.

**Response to the Editor's comments**

*Comment#1:* Comments to the author:

Dear Authors:

Thank you for your careful consideration of the referee comments and your extensive revisions in response to those comments. I think that the paper is improved - the main conclusions are easier to follow and the reasoning is presented in a reasonable layout. Given the magnitude of the changes, I sent this to one of the referees again and they agree with my overall conclusions. I am happy to accept this for publication following the attention to a few suggestions and technical corrections.

Dear Eleanor C. Browne,

Thank you very much for the supportive feedback and consideration of our paper for final publication in ACP, subject to minor comments. Our point-to-point replies to the suggestions and technical corrections are listed below:

*Comment#2:* Suggestions:

1) Please see the referee report for suggestions on modifying the conclusions to shorten the conclusions section and increase the clarity of the major findings of the work.

*Response:* Corrected in the revised manuscript and refer to author response to RC-*Comment#2*.

2) I found the results of the reducing NH3 model intriguing. It would be interesting to know if the diel profile of the NH3 in the model also more closely matched the observations as opposed to just the total amount. Given how the diel profile carries some information about processes controlling NH3 and NH4+ concentrations (which the manuscript nicely discusses earlier), I think this information would be of interest to the reader and is important information for future studies to consider. I think this could be accomplished in a few sentences - there is no need to add an extensive discussion.

*Response:* Added in the revised manuscript.

"In order to better understand the relationship between $NH_3$, $NH_4^+$ and $NH_x$ concentrations in the diurnal profile of model, one sensitivity study is conducted in the best case HCl experiment to simulate the response of $NH_x$ concentrations by changing $NH_3$ emissions. In these simulations, only $NH_3$ emissions were reduced further by a factor of 3 (-3×NH$_3$_EMI) in the 3×Base HCl experiment, while all other processes and chemical schemes were unchanged. Figure S6 in the Supplement shows the diel profile of model/obs ratio for $NH_3$

(Fig. S6a), $NH_4^+$ (Fig. S6b), and total $NH_x$ (Fig. S6c) concentration simulated with the $3\times$Base HCl and $-3\times NH_3\_EMI$ scenario. Reducing $NH_3$ emissions in the model ($-3\times NH_3\_EMI$) significantly improves model-measurement agreement for $NH_3$ (mean model/obs = 1.9), $NH_4^+$ (mean model/obs = 0.9), and total $NH_x$ concentration (mean model/obs = 1.2) compared to the $3\times$Base HCl run, further suggesting that the longer lifetime of $NH_4^+$ may be the controlling driver for the total $NH_x$ concentration in the model."

[Figure]

**Figure S6. Comparison of diel profile of model/obs ratio for the mean (a) NH₃ concentration (b) NH₄⁺ concentration, and (c) total NHₓ concentration in 3×Base HCl and -3×NH₃_EMI scenario.**

*Comment#3:* Technical

See technical corrections in Referee's report as well

*Response:* Yes, corrected in the revised manuscript.

Line 141: Suggest changing "all the precautions" to "several precautions" or "followed best practices"

*Response:* Yes, corrected in the revised manuscript.

"we have followed best practices during the study"

Line 190: Is organic carbon or total organic mass being discussed? It is unclear from the phrasing.

*Response:* Since organic carbon or total organic mass is not discussed, the statement is corrected as follows:

"This study used the Model for Ozone And Related chemical Tracers (MOZART-4) gas-phase chemical mechanism coupled with the Model for Simulating Aerosol Interactions and Chemistry (MOSAIC) aerosol scheme, that simulates $SO_4^{2-}$, $NH_4^+$, $NO_3^-$, methanesulfonate, $Na^+$, $Ca^{2+}$, $Cl^-$, carbonate, black carbon (BC), and primary organic mass (OC)."

Line 264: "ratio between observed and simulated" --> "ratio between simulated and observed" so that the wording matches the equation

*Response:* Thanks for the suggestion; corrected in the revised manuscript.

"To assess the validity of the model, the ratio between simulated and observed (model/obs) was tested."

Line 296: "aqueous phase oxidation of" --> "aqueous phase oxidation by"

*Response:* Modified in the revised manuscript.

Line 297: In the supporting information version I have, Figure S3 shows ozone and not anything directly to do with SO2/SO4. The reference to the figure is thus unclear.

*Response:* We agree with your comment; hence for clarity referencing the figure is not required, and hence we deleted it in the revised manuscript.

"In a normally $NH_3$-rich atmosphere, gas-phase oxidation of $SO_2$ is much slower than the aqueous phase oxidation of $O_3$, and due to nearby sources, much of the sulfur is present as $SO_2$ (Li et al., 2007) (Fig. S3 in the Supplement)."

Lines 320-321: Comparing the wording and the figure are confusing because the wording addresses particle phase Cl- increasing at high RH whereas the figure shows HCl/Cl- (the inverse of the wording) and thus it decreases. I suggest harmonizing the wording and the figure.

*Response:* Corrected in the revised manuscript.

"We plotted the fraction of the ratio of HCl to $Cl^-$ ($HCl/Cl^-$) as a function of $NH_4^+$ concentration and RH in Fig. 4. The decrease in the fraction of $HCl/Cl^-$ is associated with an increase in $NH_4^+$ concentration at high RH between 70-100 %."